# The biphasic and age-dependent impact of klotho on hallmarks of aging and skeletal muscle function

Zachary Clemens[1,2†], Sruthi Sivakumar[1,3†], Abish Pius[1,4†], Amrita Sahu[1], Sunita Shinde[1], Hikaru Mamiya[3], Nathaniel Luketich[3], Jian Cui[4], Purushottam Dixit[5], Joerg D Hoeck[6], Sebastian Kreuz[6], Michael Franti[6], Aaron Barchowsky[2], Fabrisia Ambrosio[1,2,3,7]*

[1]Department of Physical Medicine & Rehabilitation, University of Pittsburgh, Pittsburgh, United States; [2]Department of Environmental and Occupational Health, University of Pittsburgh, Pittsburgh, United States; [3]Department of Bioengineering, University of Pittsburgh, Pittsburgh, United States; [4]Department of Computational & Systems Biology, School of Medicine, University of Pittsburgh, Pittsburgh, United States; [5]Department of Physics, University of Florida, Gainesville, United States; [6]Department of Research Beyond Borders, Regenerative Medicine, Boehringer Ingelheim Pharmaceuticals, Inc, Rhein, Germany; [7]McGowan Institute for Regenerative Medicine, University of Pittsburgh, Pittsburgh, United States

*For correspondence:
ambrosiof@upmc.edu

[†]These authors contributed equally to this work

**Abstract** Aging is accompanied by disrupted information flow, resulting from accumulation of molecular mistakes. These mistakes ultimately give rise to debilitating disorders including skeletal muscle wasting, or sarcopenia. To derive a global metric of growing 'disorderliness' of aging muscle, we employed a statistical physics approach to estimate the state parameter, entropy, as a function of genes associated with hallmarks of aging. Escalating network entropy reached an inflection point at old age, while structural and functional alterations progressed into oldest-old age. To probe the potential for restoration of molecular 'order' and reversal of the sarcopenic phenotype, we systemically overexpressed the longevity protein, Klotho, via AAV. Klotho overexpression modulated genes representing all hallmarks of aging in old and oldest-old mice, but pathway enrichment revealed directions of changes were, for many genes, age-dependent. Functional improvements were also age-dependent. Klotho improved strength in old mice, but failed to induce benefits beyond the entropic tipping point.

## Introduction

Aging is a universal process, and decades of research have gone into understanding the cellular mechanisms underlying aged tissue phenotypes. With the goal of conceptualizing common molecular and cellular mechanisms that underlie the effect of time's arrow on mammalian tissue health, López-Otín et al. described nine 'Hallmarks of Aging' (*López-Otín et al., 2013*). These hallmarks are: epigenetic alterations, cellular senescence, altered intercellular communication, telomere attrition, nutrient sensing deregulation, mitochondrial dysfunction, stem cell exhaustion, loss of proteostasis, and genomic instability. A criterion in the identification of these hallmarks was that aggravation or attenuation of the biological process results in an accelerated aging or more youthful phenotype, respectively. These hallmarks, therefore, have the potential to pave the way toward the development of therapeutic approaches to counteract the effect of aging on organismal health.

Numerous longitudinal aging studies have positively associated skeletal muscle health with healthspan and longevity, and skeletal muscle strength has even been shown to be a reliable predictor of biological age and mortality (*Bell et al., 2019*; *Voisin et al., 2020*; *Metter et al., 2002*; *Frontera et al., 2000*). Furthermore, an age-related loss of skeletal muscle mass and function (i.e., sarcopenia) is associated with loss of mobility and increased fall risk (*Kinney, 2004*; *Landi et al., 2012*; *Imagama et al., 2011*; *Goodpaster et al., 2006*). Tissue-level features of sarcopenia have been described extensively, and include a decreased myofiber size, increased intramuscular fat accumulation, a preferential atrophy of type II (fast-twitch) muscle fibers, and compromised function (*Doherty, 2003*; *Morley et al., 2014*; *Miljkovic et al., 2015*). However, our understanding of the cellular mechanisms underlying sarcopenia is still lacking, a shortcoming that has hindered the development of targeted and specific interventions (*Marzetti et al., 2017*; *Dhillon and Hasni, 2017*; *Woo, 2017*; *Wang et al., 2019*). Currently, approaches to the treatment and prevention of sarcopenia largely focus on the prescription of physical activity and dietary modifications, strategies that have shown moderate success (*Morley, 2018*; *Waters et al., 2010*; *Taaffe, 2006*). Furthermore, proven pharmacological interventions for sarcopenia do not exist, although drugs such as vitamin D, insulin-like growth factors, testosterone, cetylpyridinium chloride (CPC2), as well as monoclonal antibody treatments have entered advanced clinical trials (*Woo, 2017*; *Drescher et al., 2016*; *Rolland et al., 2011*; *Becker et al., 2015*; *Bauer et al., 2015*; https://clinicaltrials.gov/ct2/show/NCT01963598, https://clinicaltrials.gov/ct2/show/NCT02333331, https://clinicaltrials.gov/ct2/show/NCT02575235). Interestingly, many of the drugs currently being tested also act on pathways associated with the longevity protein, α-Klotho (Klotho).

Klotho is best known for its role in delaying age-related pathologies in various organ systems, including skeletal muscle (*Kurosu et al., 2005*; *Kuro-o et al., 1997*). The pro-longevity effects of Klotho have been partially attributed to modulation of fibroblast growth factor 23 (FGF23), Wnt, and mTOR pathways (*Hu et al., 2013*; *Liu et al., 2007*; *Zhao et al., 2015*), several of which have also been a targets in sarcopenia research (*Woo, 2017*; *Becker et al., 2015*; *Yoshida et al., 2019*; *D'Antona and Nisoli, 2010*). With age, circulating Klotho levels gradually decline, and epidemiological studies have revealed that decreased circulating Klotho levels are associated with an accelerated loss of skeletal muscle mass and strength (*Yamazaki et al., 2010*; *Semba et al., 2016*). Similarly, mice deficient for Klotho display significant muscle wasting, which is hypothesized to be caused by increased mitochondrial reactive oxygen species (ROS) (*Sahu et al., 2018*; *Phelps et al., 2013*). These findings are consistent with the observation that mitochondrial accumulation of ROS leads to a decline in muscle mass and decreased regenerative potential (*Sahu et al., 2018*; *Leduc-Gaudet et al., 2015*). Taken together, these studies suggest an unexplored mechanistic link between sarcopenia and a decline of Klotho with increasing age.

In this study, we first thoroughly characterize and compare structural, functional, and transcriptomic changes in skeletal muscle across the lifespan in mice. In order to derive a global metric of the loss of molecular fidelity over time, we used an information-based calculation of network entropy from muscle RNA-seq data. A higher network entropy means a probabilistically greater degree of disorganization, or randomness, in the system. We found that network entropy increases from young (4–6 months) to old (21–24 months) mice, but then plateaus in the oldest-old mice (27–29 months). We then tested the ability of AAV-mediated Klotho delivery (AAV-Kl) to attenuate these changes and counteract sarcopenia in old and oldest-old mice. The impact of AAV-Kl was age-dependent, and whereas old mice displayed a significant improvement in function, AAV-Kl failed to induce a benefit in the oldest-old mice. Furthermore, AAV-Kl regulated a multitude of genes associated with all hallmarks of aging. Unexpectedly, however, the direction of change in response to treatment was, for many genes, dependent on the age of the host. Taken together, these results suggest Klotho enhances skeletal muscle structure and function at the early stages of sarcopenia, but the beneficial effects are lost after the entropic inflection point.

## Results

### Sarcopenic changes are subtle until mice reach an advanced age

To thoroughly describe the trajectory of sarcopenic alterations according to sex and over time, we characterized muscle structure and function in young (3–6 months), middle-aged (10–14 months),

old (21–24 months), and oldest-old (27–29 months) male and female C57BL/6J mice. Age groups were selected to parallel stratifications commonly used in epidemiological studies and correspond, roughly, to individuals aged 20–30, 38–47, 56–69, and 78 + years (*Dutta and Sengupta, 2016*; *Cruz-Jentoft et al., 2019*; *Sayer et al., 2013*). As expected, a progressive decline in tibialis anterior (TA) wet weight (normalized to body weight) was observed over time for both males and females, though the percent decrease at the oldest-old age was significantly greater in male mice (34%) when compared to females (17%; *Figure 1A*). A decline in TA weight was concomitant with a decline in physiological cross-sectional area (CSA, *Figure 1—figure supplement 1A*). Bulk decreases in size and cross-sectional area were accompanied by a decrease in individual myofiber cross-sectional area for both sexes, though declines were greater in female mice (*Figure 1B,C*, *Figure 1—figure supplement 1C*). The total number of muscle fibers did not change with age (*Figure 1—figure supplement 1B*).

Myofiber typing was quantified by staining for myosin heavy chain markers of the dominant fiber types in mouse TA muscles. A decrease in the percentage of the more oxidative Type IIA and IIX muscle fibers was observed in the oldest-old, but not old, age group for both sexes (*Figure 1C,D*, *Figure 1—figure supplement 1C*). These changes were accompanied by an increase in the percentage of the glycolytic Type IIB phenotype and a decrease in fiber area of all fiber types in oldest-old groups (*Figure 1C*, *Figure 1—figure supplement 1C–G*). These results are consistent with a loss of oxidative fiber types with age observed in previous murine studies (*Augusto et al., 2017*; *Giacomello et al., 2020*). We also observed increased intramuscular collagen deposition in the oldest-old age groups of both sexes (*Figure 1C,E*, *Figure 1—figure supplement 1C*). Unexpectedly, there was no age-related increase in intermuscular lipid accumulation in either sex (*Figure 1C*; *Figure 1F*). This contrasts with clinical reports of muscle quality in older individuals (*Doherty, 2003*; *Morley et al., 2014*). We did observe a small increase in intramuscular lipid only in old male mice (*Figure 1G*). Finally, given that denervation commonly accompanies a sarcopenic profile in clinical populations, we probed transcript levels of common markers of muscle denervation (*Musk*, *Ncam1*, *Runx1*) (*Aare et al., 2016*) in young, middle-aged, old, and oldest-old skeletal muscle RNA-seq data that is publicly available through the Tabula Muris Senis (TMS) database (*Schaum et al., 2020*). *Musk* and *Runx1* displayed increased expression in oldest-old mice, although sex differences could not be determined due to inadequate power (*Figure 1—figure supplement 2*).

Whole-body muscle strength, as measured by the four-limb hang test, gradually declined with increasing age for both male and female mice (*Figure 2A*). However, in vivo contractile testing of the TA muscle revealed a significant decline in muscle strength only in the oldest-old mice, regardless of sex (*Figure 2B,C,D*). Changes in temporal characteristics of the muscle contraction profile, including increased time to peak twitch contraction and half relaxation time, were also most prominent in the oldest-old female group (*Figure 2E,F*). These temporal changes are consistent with the decreased area of fast-twitch fiber phenotypes and coincide with clinical reports (*Miljkovic et al., 2015*; *Larsson et al., 1978*; *Lexell and Taylor, 1991*). Taken together, these data suggest that old mice display a mild sarcopenic profile and that common features associated with clinical metrics of sarcopenia only become evident in oldest-old mice.

## Aging drives a progressive disruption in genes associated with hallmarks of aging

Structural and functional findings across age groups revealed exaggerated phenotypic changes between old and oldest-old female mice. To investigate corresponding molecular-level changes in the muscles of young, old, and oldest-old female hindlimb muscle, we performed RNA-seq analysis from gastrocnemius muscle lysates (*Figure 3A*). We excluded middle-aged muscles from the analyses since this age group was phenotypically similar to old mice. Principal Component Analysis (PCA) revealed distinct gene expression profiles according to age, with young muscle segregating from the old and oldest-old muscle profiles, the latter of which slightly overlapped (*Figure 3B*). These findings are consistent with previous transcriptomic studies of human skeletal muscle in which the bulk of gene expression changes occur between young and old age, after which time they plateau (*Cellerino and Ori, 2017*; *Byrne et al., 2019*).

Given that distinction across age clusters was most evident for PC2, we probed for the top 100 genes in PC2 that varied according to age. Of these, 59 out of 100 genes were associated with hallmarks of aging and, most prominently, genes associated with altered intercellular communication

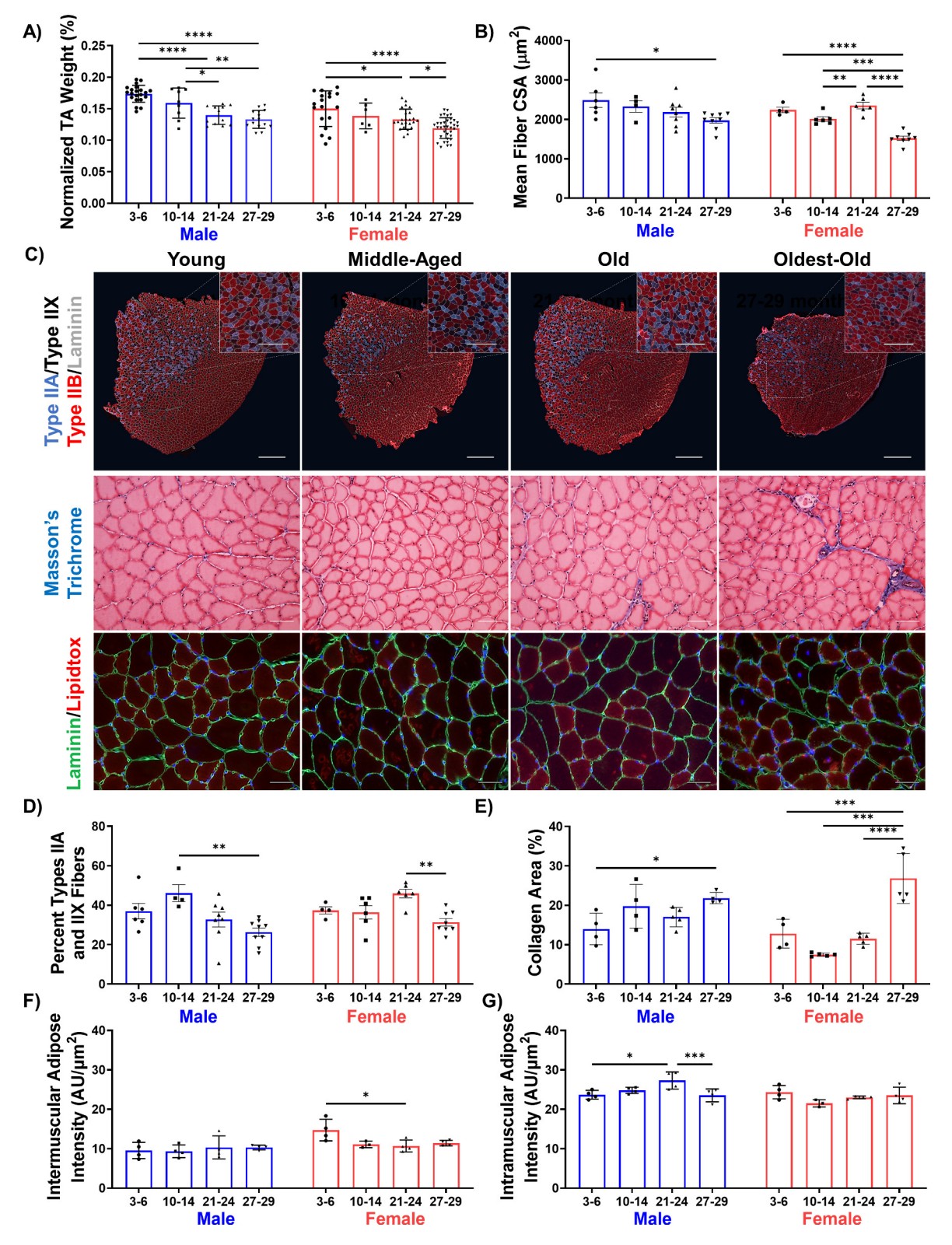

**Figure 1.** Declines in muscle structure are subtle until advanced age. (**A**) Tibialis anterior (TA) muscle weight as a percentage of whole-body weight in young (3–6 months), middle-aged (10-14 months), old (21-24 months), and oldest-old (27-29 months) male (N = 57) and female mice (N = 78, one-way ANOVAs). (**B**) Average fiber cross-sectional area of uninjured male (N = 27) and female (N = 24) mouse TAs across age groups (one-way ANOVAs). (**C**) Representative images of TA sections stained for laminin (gray), type IIA (purple), type IIX (black/unstained), and type IIB (red) fibers (top, main scale

*Figure 1 continued on next page*

*Figure 1 continued*

bars = 500 μm, inset scale bars = 250 μm ); Masson's trichrome staining (middle, 50 μm); and lipidtox staining (bottom, lipidtox = red, laminin = green, scale bars = 50 μm). (**D**) Percentage of IIA and IIX fibers in the whole TA cross-section of male (N = 27) and female (N = 24) mice (one-way ANOVAs). (**E**) Collagen area of TA sections across ages and sexes (male N = 17, female N = 19) quantified by Masson's Trichrome staining (one-way ANOVAs). (**F**) Intermuscular lipid accumulation in the TA across ages and sexes (male N = 16, female N = 15) quantified by lipidtox staining (one-way ANOVA). (**G**) Intramuscular lipid accumulation in the TA across ages and sexes (male N = 16, female N = 15) quantified by lipidtox staining (one-way ANOVA). All data presented as mean ± SD (*p<0.05, **p<0.01, ***p<0.001, ****p<0.0001).

The online version of this article includes the following source data and figure supplement(s) for figure 1:

**Source data 1.** Raw Data for *Figure 1A-B, D-G*, and *Figure 1—figure supplement 1A-B, D-G* and *Figure 1—figure supplement 2*.
**Figure supplement 1.** Characterization of sarcopenic changes in male mice.
**Figure supplement 2.** Denervation gene expression in mice across age groups for *Musk* (N = 28), *Ncam1* (N = 27), and *Runx1* (N = 29).

and nutrient-sensing deregulation. The heatmap shows some PC2 genes displayed a steady trend with aging progression (*Figure 3C*). Gene Ontology enrichment confirmed that the top 100 genes for PC2 are associated with cellular homeostasis, cellular structure, and intercellular communication (*Figure 3—figure supplement 1A,B*).

To date, most RNA-seq studies in skeletal muscle have used a bottom-up approach, focusing on differentially expressed (DE) genes and their corresponding pathways. Whereas this approach is valuable for identifying specific signaling pathways that may contribute to sarcopenia, it does not give an integrative view of underlying pathogenic mechanisms. Therefore, we employed a top-down approach, whereby we constructed interactomes according to muscle transcriptional profiles over time. It has been proposed that 'the common denominator that underlies all modern theories of aging is change in molecular structure and, hence, function.' (*Hayflick, 2007*) Although the entropy theory of aging has existed for centuries, entropy has generally been considered in the context of thermodynamics (i.e., 'dissipation' of energy at the biomolecular level). While theoretically intriguing, this form of entropy is challenging to quantify. Given that the essence of entropy is a loss of information in a system, we estimated network entropy as described by *Menichetti et al., 2015*; *Menichetti and Remondini, 2014*. To do this, protein-protein interaction (PPI) networks were generated from RNA-seq data across young, old, and oldest-old age groups. Given our focus on age-related alterations, only those genes associated with the hallmarks of aging were included in the analysis (*López-Otín et al., 2013*). In the network, nodes represent the corresponding proteins, edges represent the physical interaction of proteins, and edge weights represent the differential gene expression (*Figure 3D*). Genes associated with 'stem-cell exhaustion' largely overlapped with all other hallmarks, and, hence, were excluded from our analysis.

Consistent with PCA findings, but contrary to phenotypic changes, we observed the greatest increase in network entropy between young and old groups, with little difference between the old and the oldest-old (*Figure 3E*). This suggests that gene expression changes likely precede the observed functional deficits in oldest-old mice (*Figure 2*). The same entropy trend emerged even when all genes were considered, suggesting that the subset of genes associated with the hallmarks of aging captures the overall tissue transcriptomic profile without loss of information (*Figure 3—figure supplement 2*). Furthermore, we found that the increase in entropy between young and old muscles was driven by a relatively small pool of genes that displayed a large magnitude of change (*Figure 3—figure supplement 3*). In contrast, there were many more DE genes when comparing old and oldest-old muscles, although the magnitudes of change were generally low (*Figure 3—figure supplement 3*). This suggests a loss of specific changes and a broadening of gene expression alterations across the genome in the oldest-old age group. We cross-checked entropy trends in skeletal muscle of young, middle-aged, old, and oldest-old male mice using RNA-seq data available through the TMS database (*Schaum et al., 2020*). Unlike females, we saw that entropy continued to increase from old to oldest-old timepoints (*Figure 3—figure supplement 4A*), which we found interesting considering that male mice showed less severe sarcopenic declines at the oldest-old age (*Figures 1* and *2*). Also interesting, entropy was lowest at the middle-aged timepoint, then gradually increased.

To probe the translational potential of the network entropy approach, we used open-source RNA-seq data from aging male rats and humans (SRA: PRJNA516151 and GSE164471) (*Shavlakadze et al., 2019*; *Tumasian et al., 2021*). We did not present female data for either species due to a lack of sufficient sample sizes. In rats, entropy decreased from young to middle-age,

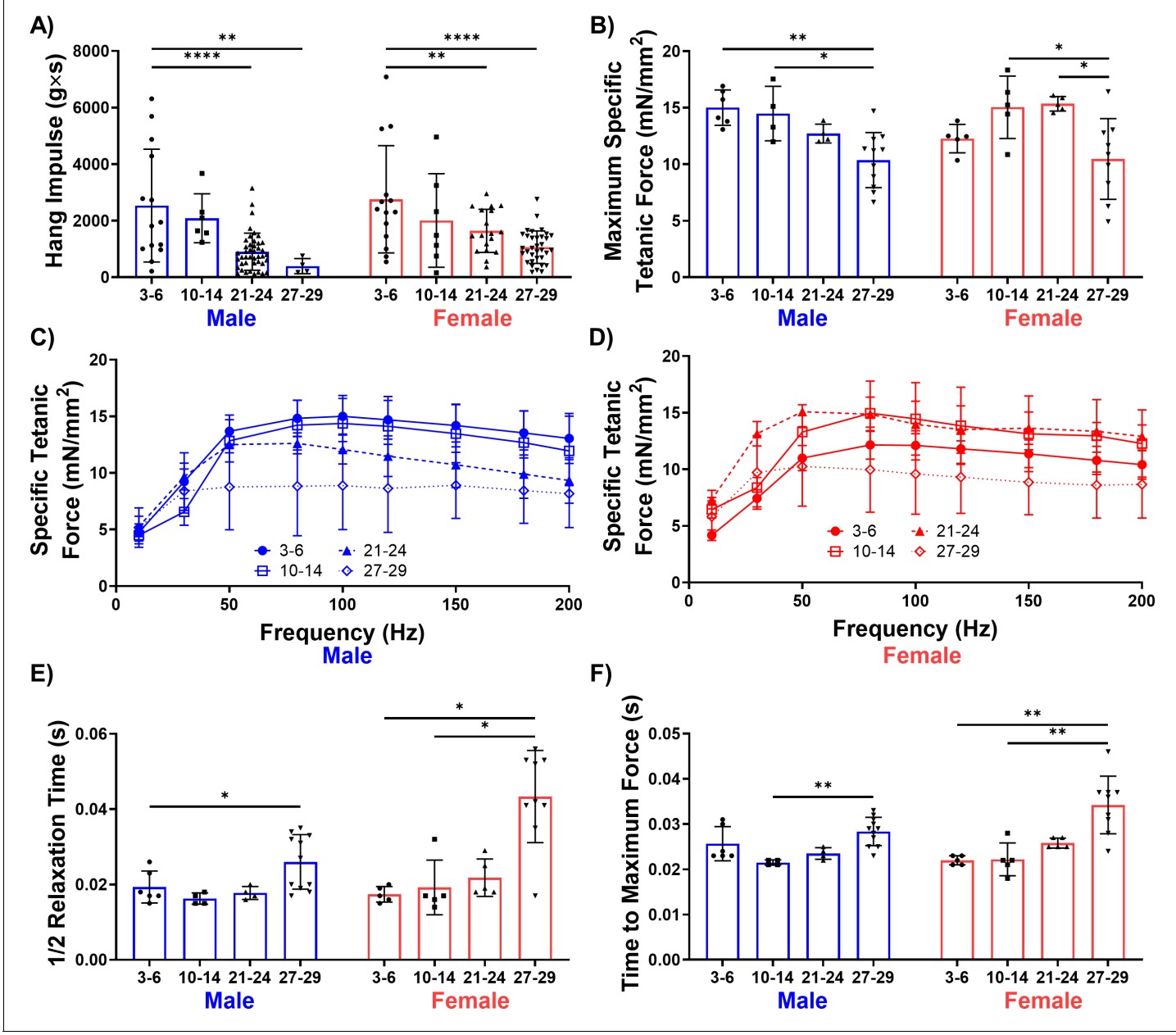

**Figure 2.** Male and female mice display a progressive loss of muscle function over time. (A) Whole-body endurance of young (3–6 months), middle-aged (10-14 months), old (21-24 months), and oldest-old (27-29 months) male (N = 67) and female (N = 77) mice measured by the four-limb hang test (one-way ANOVA). 'g'= gravity and 's'=seconds. (B) Peak specific tetanic force production in male (N = 25) and female (N = 24) TA muscles (one-way ANOVAs). (C) Force frequency curves for TA stimulation in male mice (N = 25). (D) Force frequency curves for TA stimulation for female (N = 24) mice. (E) Half relaxation time of the TA muscle following single twitch stimulation across ages and sexes (male N = 25, female N = 24, Kruskal-Wallis tests). (F) Time to maximum force following single twitch stimulation of the TA muscle (male N = 25, female N = 24, Kruskal-Wallis tests). All data presented as mean ± SD (*p<0.05, **p<0.01, ****p<0.0001).

The online version of this article includes the following source data for figure 2:

**Source data 1.** Raw Data for *Figure 2*.

after which time it increased into old age, consistent with the aforementioned trend observed in male mice (*Figure 3—figure supplement 4B*). Oldest-old muscles were not available in rats. Entropy in humans similarly decreased from young adulthood into the fourth decade of life, after which time it increased, reaching a tipping point in the sixth decade of life, then decreasing slightly (*Figure 3—figure supplement 4C*). It is interesting to note that the entropic nadir in humans coincides with the

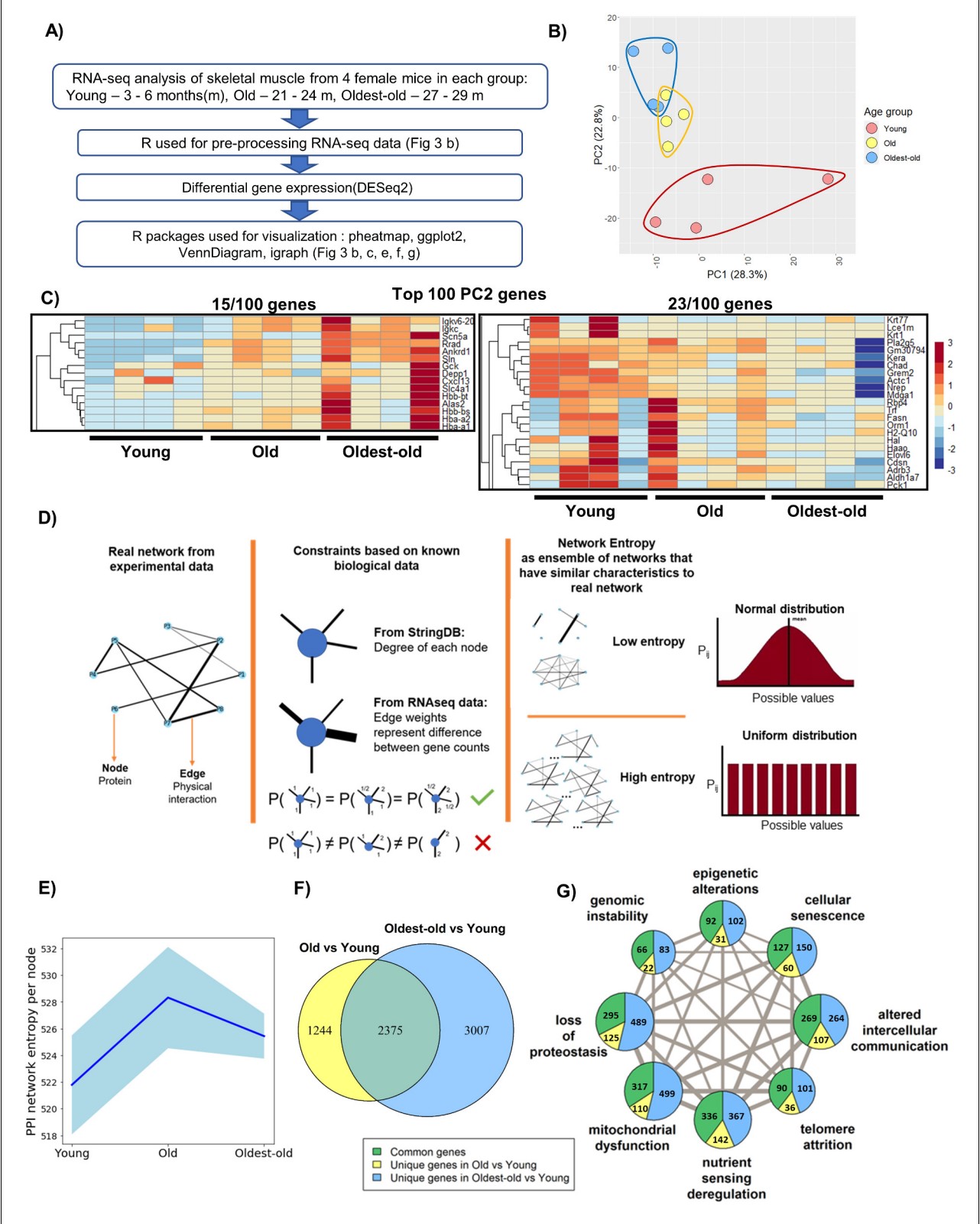

**Figure 3.** Network entropy increases from young to old mice, after which time it plateaus. (A) RNA-seq analysis workflow. (B) Principle component analysis (PCA) showing overall gene expression patterns in young (3–6 months), old (21-24 months), and oldest-old (27-29 months) mice (N = 12). (C) Heatmap showing genes associated with aging progression derived from the 100 genes with the highest PC2 loadings. The two highlighted sections show 15/100 genes that have an increasing trend and 23/100 genes that have a decreasing trend. (D) Schematic description of network entropy

*Figure 3 continued on next page*

*Figure 3 continued*

computation and interpretation. PPI networks were generated based on RNA-seq data. We capture the degree sequence and the edge weights from the network obtained from experimental data in the form of constraints. The ensemble of networks that follow these constraints have similar network features. If the probability distribution is skewed, then it has a low network entropy and, if not, then it is has a high network entropy. (E) Protein-protein interaction (PPI) network entropy computed from transcriptomic data indicates an increase in molecular disorder of hallmarks of aging genes. A non-parametric Kruskal Wallis test (p=0.0741) and Dunn's post-hoc test were performed. Entropy of young to old changed with p=0.07. Blue shaded portion indicates standard deviation (n = 4). (F) Venn Diagram showing the number of differentially expressed (DE) genes between groups old vs. young, and oldest-old vs. young mice. (G) Network plot denotes the total number of DE genes per hallmark and the corresponding proportion of DE genes that are unique to old and oldest-old, when compared to young counterparts for each hallmark of aging. Edge weights denote the number of genes that are common between the two hallmarks the edge connects. The node sizes are proportional to the number of genes that fall into each hallmark.

The online version of this article includes the following source data and figure supplement(s) for figure 3:

**Source data 1.** Raw Data for *Figure 3—figure supplement 4*.
**Figure supplement 1.** Transcriptomic changes in the context of hallmarks of aging genes.
**Figure supplement 2.** Network entropy with all genes.
**Figure supplement 3.** Histogram of all gene counts across young, old, and oldest-old age groups.
**Figure supplement 4.** Validation of network entropy trends in other samples and species.

commonly observed onset of sarcopenic declines in the fourth decade of life (*Walston, 2012*; *Yazar and Olgun Yazar, 2019*). With the goal of providing the broader research community an easily accessible tool for estimating network entropy from transcriptomic and proteomic datasets, we created an open-source Python streamlit web-app (https://network-entropy-calculator.herokuapp.com/). Preprocessing steps are detailed on the Github link (https://github.com/sruthi-hub/sarcopenia-network-entropy).

We next probed for 'weakest links' among the hallmarks of aging in which the effect of aging may become manifest prior to the others. Unexpectedly, we found that all of the hallmarks were similarly affected when considering changes between young and old mice and young and the oldest-old mice, although roughly twice as many genes changed in the oldest-old mice (*Figure 3F,G*). This further implies a dysregulation of the transcriptomic profile in the oldest-old mice, a finding that is consistent with previous reports (*Cellerino and Ori, 2017*).

## AAV-Klotho administration improves muscle regeneration after severe, but not exercise-induced, injury in old mice

Given that a criterion in the identification of aging hallmarks is a susceptibility to interventions that may aggravate or attenuate age-related declines, we next tested whether the declines in muscle function and structure observed in aged mice could be attenuated through Klotho supplementation. As a first step, we confirmed that circulating Klotho gradually declines with age in mice, consistent with human findings (*Yamazaki et al., 2010*; *Figure 4A*). In both sexes, we also observed a stepwise increase in circulating fibroblast growth factor-23 (FGF23), a known co-factor for Klotho signaling and a key hormone in the regulation of mineral ions and vitamin D (*Chen et al., 2018*; *Figure 4B*). It was interesting to note that circulating FGF23 levels in the oldest-old females clustered into two different groups, with approximately 40% of the population displaying very high levels compared to the more linearly increasing levels of the remaining mice. This variation may be a result of varied hormonal status across the cohort. Although hormone levels were not measured here, it has previously been shown that 20–40% of female mice spontaneously regenerate follicles after menopause (*Diaz Brinton, 2012*), which is relevant given that estrogen levels have been shown to regulate circulating FGF23 in both rodents and humans (*Ix et al., 2011*; *Saki et al., 2020*).

Next, an AAV vector carrying a liver-specific promoter and mouse Klotho full-length cDNA (AAV-Kl) was constructed and administered via tail vein injection (*Figure 4C*). Three weeks after administration, qPCR confirmed that the vector was inserted into the genome in the liver and replicated (*Figure 4D*). Meso Scale Discovery-Enzyme-linked Immune Sorbent Assay (MSD-ELISA) conducted on serum further confirmed a dose-dependent increase in circulating Klotho levels (*Figure 4E*). Klotho transcript levels in the gastrocnemius muscles of old and oldest-old mice were unaffected by AAV-Kl administration, consistent with use of a liver-specific promoter (*Figure 4F*). Lastly, the AAV expression of Klotho had no significant effects on non-fasting serum insulin, glucose, cholesterol, or other circulating lipids (*Figure 4G,H,I*; *Figure 4—figure supplement 1*).

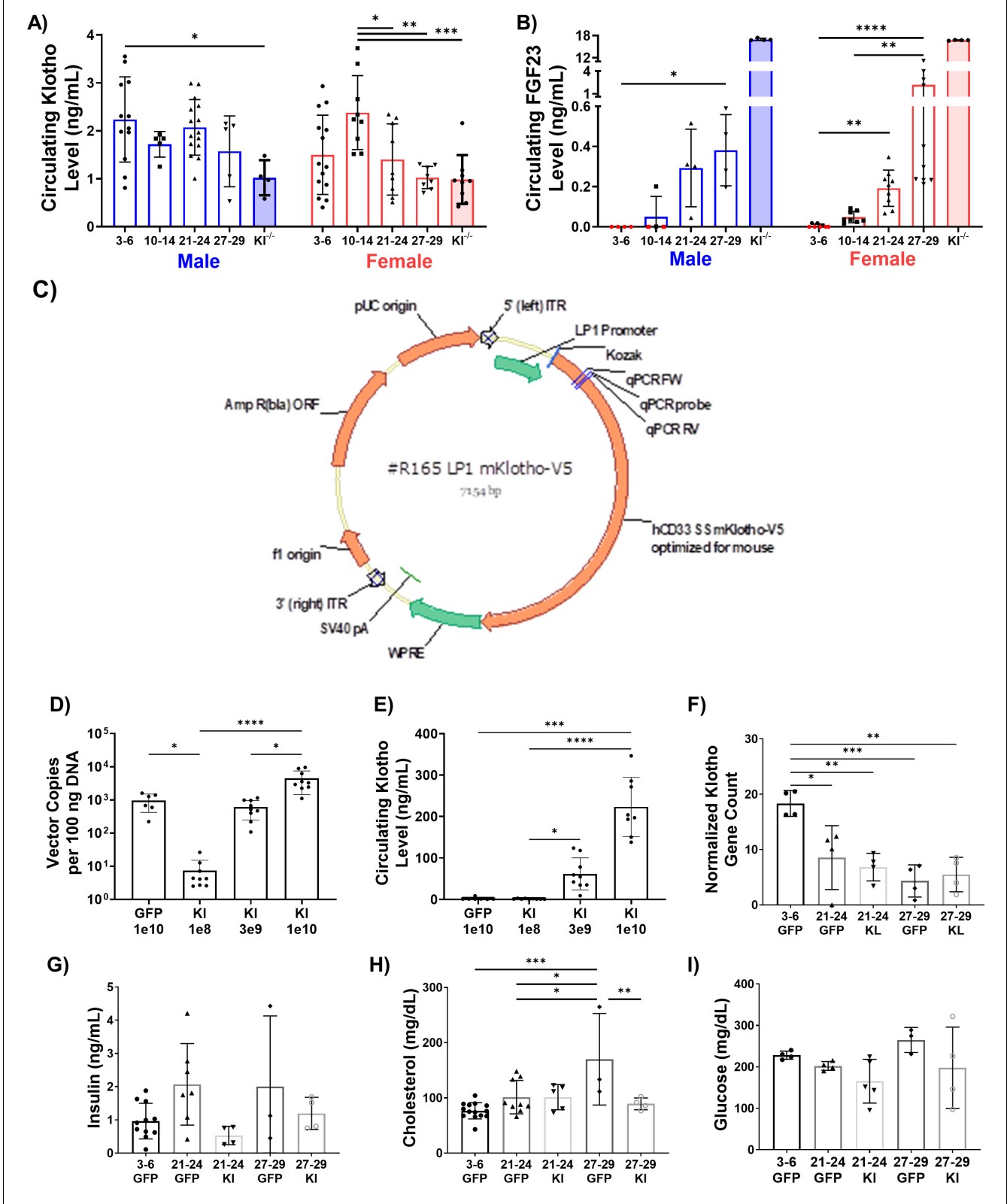

**Figure 4.** Development and validation of an AAV approach for systemic delivery of Klotho. (**A**) Changes in circulating Klotho levels measured via ELISA in young (3–6 months), middle-aged (10-14 months), old (21-24 months), and oldest-old (27-29 months) male (N = 41), and female (N = 47) mice (one-way ANOVAs). (**B**) Changes in circulating FGF23 levels in male (N = 20) and female (N = 39) mice. Red symbols represent undetectable levels and were set to zero (Kruskal-Wallis tests, KO values were excluded from statistical analysis). (**C**) Schematic of the AAV-Klotho plasmid design. (**D**) Liver expression

*Figure 4 continued on next page*

Figure 4 continued

of AAV vector genomes quantified via qPCR (N = 33, Kruskal-Wallis test). (E) Circulating Klotho levels measured via MSD-ELISA in young female (N = 33) mice injected with AAV-Klotho at varying doses (Kruskal-Wallis test). (F) Gene count normalized to library size for Klotho in the gastrocnemius muscle of female mice treated with GFP and AAV-Kl (N = 20, one-way ANOVA). (G,H,I) Serum concentration levels for insulin (N = 29), cholesterol (N = 35), and glucose (N = 20) in GFP- and Kl-treated female mice (one-way ANOVA). All data presented as mean ± SD (*p<0.05, **p<0.01, ***p<0.001, ****p<0.0001).

The online version of this article includes the following source data and figure supplement(s) for figure 4:

**Source data 1.** Raw Data for *Figure 4A-B, D-I*, and *Figure 4—figure supplement 1*.
**Figure supplement 1.** Quantification of circulating lipid metabolites in mice receiving AAV-Kl treatment versus controls.

As a means to confirm the physiological effect of AAV-Kl administration, we tested whether AAV-Kl administration could replicate data from a previous report demonstrating improved regeneration following supplementation with recombinant Klotho protein (*Sahu et al., 2018*). AAV-Kl (1 × 10^10 vector genomes/animal) or an equal titer AAV-GFP control was administered via tail vein injections to old male mice 5 days before bilateral injuries to the tibialis anterior (TA) muscles with cardiotoxin. Histological analysis and in vivo contractile testing was performed 14 days after injury (19 days after AAV injection) (*Figure 5A*). There was no difference in myofiber cross-sectional area in the injured TAs of animals treated with AAV-Kl when compared to control counterparts (*Figure 5B*). However, fibrosis was significantly decreased in the TAs of AAV-Kl-treated mice, as evidenced by decreased collagen deposition (*Figure 5C,D,E*). This decrease in fibrosis was concomitant with a decreased number of fibroadipogenic progenitor cells (FAPs) with AAV-Kl treatment, as quantified by flow cytometry and following a previously published protocol (*Yi and Rossi, 2011*; *Liu et al., 2015*; *Figure 5F*). In the light of previous work demonstrating that Klotho supplementation enhances mitochondrial ultrastructure in aged myogenic cells (*Sahu et al., 2018*), we also performed TEM and compared mitochondrial integrity in muscles across experimental groups. Indeed, AAV-Kl administration significantly increased the percentage of healthy mitochondria in aged regenerating muscle (*Figure 5G,H*).

Structural improvements were consistent with functional outcomes, and mice receiving AAV-Kl also displayed increased specific twitch and maximum specific tetanic force production (*Figure 5I,J*). Resistance to a fatiguing protocol was not affected by treatment, although force recovery following completion of the fatiguing protocol was significantly enhanced in mice treated with AAV-Kl (*Figure 5K*). Additionally, whole-body strength improved almost twofold over 14 days for mice treated with AAV-Kl; control mice showed no improvement (*Figure 5L*). These results confirmed that systemically elevated Klotho levels enhance skeletal muscle regeneration, similar to a previous report using intraperitoneal injection of recombinant Klotho (*Sahu et al., 2018*).

To assess the efficacy of Klotho overexpression in a second, less severe, injury model, we evaluated the ability of AAV-Kl administration to enhance recovery following an exercise-induced injury. To this end, we conducted an eccentric exercise protocol adopted from *Armand et al., 2003*; *Figure 5—figure supplement 1A,B*. This downhill running model has been shown to induce eccentric muscle injury in tibialis anterior and soleus muscles (*Armand et al., 2003*; *Parise et al., 2008*; *Breen and Phillips, 2011*). Briefly, old male mice that received either AAV-Kl or GFP, as above, were acclimated to treadmill running for 4 days. On the fifth day, mice were placed on a treadmill to run for 60 min at 10 m/min on a 15-degree decline. We assessed exercise adherence according to a previous report in order to confirm that animals in both groups were similarly able to complete the protocol (*Ríos-Kristjánsson et al., 2019*; *Figure 5—figure supplement 1C*). Mice with an adherence score of less than 0.30 were excluded from the analysis (n = 2 per group; *Figure 5—figure supplement 1C*). We also confirmed that the protocol induced TA muscle injury using histological analysis (*Figure 5—figure supplement 1D,E*). In vivo contractile testing performed 7 days after injury revealed that animals treated with AAV-Kl displayed no significant increase in muscle performance (*Figure 5M*).

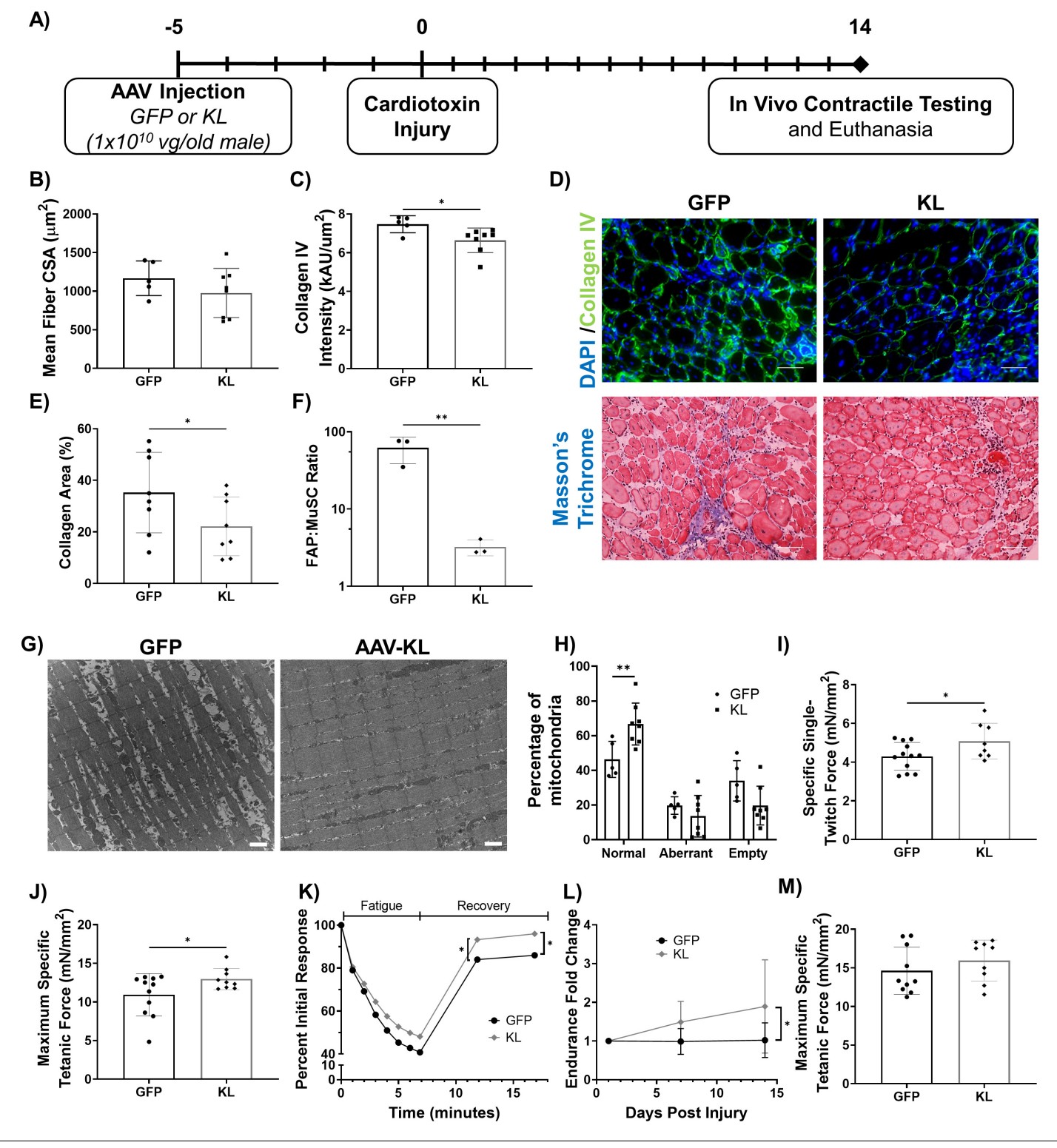

**Figure 5.** Gene delivery of Klotho enhances functional muscle regeneration following an acute injury. (**A**) Experimental design using old (21–24 months) male mice. (**B**) Quantification of TA average myofiber cross-sectional area (N = 13). (**C**) Collagen IV expression in the TA muscle of GFP- versus KL-treated mice (one-tailed Mann-Whitney test, N = 13). (**D**) Top: Representative images of injured TA muscles stained for collagen IV (green) and DAPI (blue, scale bars = 50 μm). Bottom: Masson's Trichrome staining of the TA (scale bars = 50 μm). (**E**) Collagen area percentage in the TA quantified from Masson's Trichrome staining (one-tailed student's t-test, N = 16). (**F**) FAPs to MuSCs ratio in injured TA muscles, as determined by flow cytometry (N = 6, one-tailed student's t-test). (**G**) Representative TEM images showing mitochondria in the TA muscle fibers of AAV-GFP vs. KL-treated mice.

*Figure 5 continued on next page*

*Figure 5 continued*

Aberrant and empty mitochondria show abnormal shape and high proportion of white space respectively (scale bars = 1µm). (H) Quantification of the quality of mitochondria (two-way ANOVA, N = 13). (I) TA specific twitch force produced 14 days post-injury (dpi) (one-tailed Student's t-test, N = 20). (J) TA maximum specific tetanic force 14 dpi (one-tailed Student's t-test, N = 20). (K) Change in force production of the TA over time as mice underwent a fatigue protocol consisting of repeated TA stimulation for a total of 7 min, followed by recovery over two 5-min intervals (two-way ANOVA, N = 19). (L) Fold change in whole body endurance compared to one day post injury hang impulse score (Mixed-effects analysis, N = 16). (M) TA peak tetanic specific force for mice 7 days after an eccentric injury treadmill protocol (N = 19). All data presented as mean ± SD (*p<0.05, **p<0.01). The online version of this article includes the following source data and figure supplement(s) for figure 5:

**Source data 1.** Raw Data for *Figure 5B-C, E-F, H-M*, and *Figure 5—figure supplement 1C, E*.
**Figure supplement 1.** Validation of the eccentric exercise injury model.

## Intravenous AAV-Klotho administration enhances muscle structure and function in old, but not in the oldest-old, mice

We next asked whether AAV-Kl administration could reverse age-related declines in muscle quality and function even in the absence of injury. Female mice received either AAV-Kl or AAV-GFP at a titer of $3 \times 10^8$ vg/mouse via tail vein injection. This dose was chosen because it provided the greatest improvement in muscle strength, as determined over the course of a preliminary dose-testing study performed in old mice (*Figure 6—figure supplement 1A*). Fourteen days after injection, mice were euthanized and endpoints collected (*Figure 6A*). A challenge of studying sarcopenia in mouse models is the high rate of mortality and the onset of confounding pathologies in very old animals. Of the 51 old and oldest-old mice included, only 30 mice were included in final analyses due to incidental morbidity and mortality, with most of these losses occurring prior to randomization (*Figure 6B*).

Histological analysis of old muscles revealed no improvement in the muscle weight, myofiber cross-sectional area, fiber number, or percentage of type IIA/X fibers with AAV-Kl treatment (*Figure 6C,D,E,F*, *Figure 6—figure supplement 1B*). There was also no change in expression of the previously mentioned denervation genes with Klotho treatment (*Figure 6—figure supplement 1C*). However, intramuscular lipid accumulation was significantly decreased following treatment (*Figure 6C,G*). AAV-Kl enhanced muscle function in old female mice. Specifically, maximum specific force production improved by 17% (*Figure 6H*), and whole-body endurance measured via four-limb hang test was 60% greater when compared to control counterparts (*Figure 6I*). Encouraged by the results in old mice, we then administered AAV-Kl or AAV-GFP to a cohort of oldest-old female mice, which displayed a more severe sarcopenic phenotype (*Figures 1* and *2*). Unlike old mice, there was no improvement in any of the metrics evaluated (*Figure 6C,J–O*, *Figure 6—figure supplement 1B, D*). In fact, AAV-Kl administration significantly *increased* overall lipid accumulation (*Figure 6C,M*). These findings suggest that systemic upregulation of Klotho induces therapeutic benefits in old mice, but not in mice of more advanced age.

## Klotho affects hallmark of aging genes across age groups, but oldest-old mice exhibit a dysregulated response

To understand the benefit of Klotho in the old group and lack of benefit in the oldest-old group, we revisited the transcriptomic profiles across experimental groups according to the hallmarks of aging. When comparing age-matched control and AAV-Kl treatment groups, we observed approximately three times more DE genes in the oldest-old group treated with Klotho as compared to the old group (*Figure 7A*). Existing literature suggests that Klotho attenuates hallmarks associated with mitochondrial dysfunction, telomere attrition, and cellular senescence (*Sahu et al., 2018*; *Kuro-o, 2008*; *Ullah and Sun, 2019*; *Frenk and Houseley, 2018*). However, we found that AAV-Kl affected all the hallmarks similarly (*Figure 7B*).

To improve resolution between the hallmarks, we probed for pathways associated with the 473 and 1419 DE genes in Klotho-treated old and oldest-old mice relative to their age-matched controls. DE genes responsive to Klotho therapy in old mice were associated with pathways related to inhibition of FGFR4/calcineurin/NFAT signaling. These findings are consistent with previous reports demonstrating that overexpression of Klotho reverts pathogenic FGF23 signaling to more normal tissue homeostasis (*Xiao et al., 2019*; *Ho and Bergwitz, 2021*). Intriguingly, however, FGFR4/calcineurin/NFAT signaling did not rise to the top of gene expression changes in response to Klotho treatment

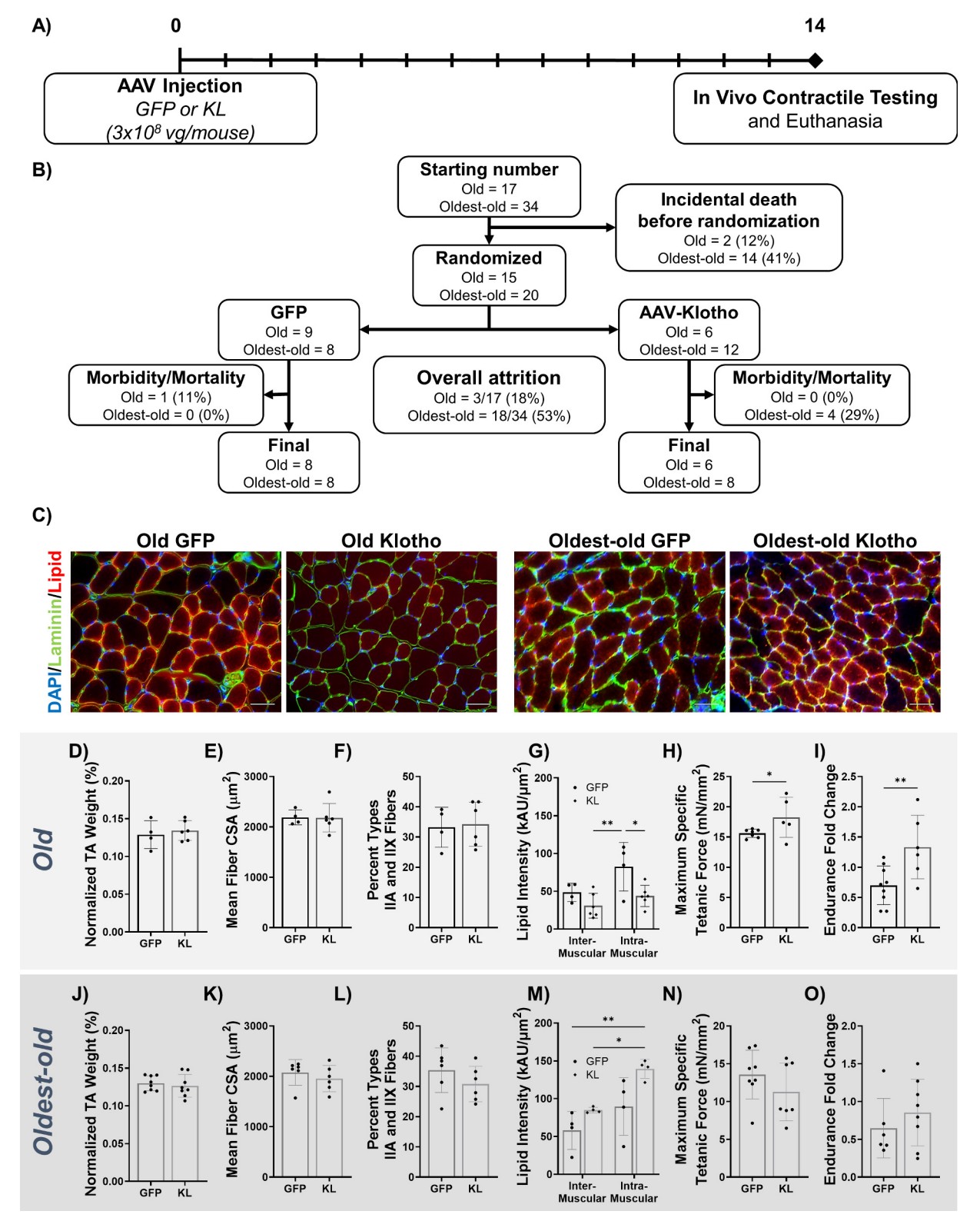

**Figure 6.** AAV-Klotho enhances muscle function in old, but not oldest-old, mice. (**A**) Experimental design and timeline using old (21–24 months) and oldest-old (27-29 months) female mice. (**B**) Animal inclusion flow chart. Mortality describes mice that died over the course of the experiment. Morbidity describes mice in whom pathology was found at the time of euthanasia . These mice were subsequently excluded from analyses. (**C**) Representative images showing TA myofiber area (Laminin; green), lipid (red), and DAPI (blue) of the TA 14 dpi in old and oldest-old female mice treated with GFP or

*Figure 6 continued on next page*

Figure 6 continued

Kl. Scale bars = 50 µm. (D) TA wet weight (as a percent of total body weight) of old female mice treated with AAV-GFP or AAV-Kl (N = 10). (E) Quantification of TA muscle-fiber cross-sectional area (CSA) for old female mice (N = 10). (F) Percentage of type IIA and IIX muscle fibers in whole TA cross-sections of old female mice (N = 10). (G) Inter- and intramuscular lipid intensity in TA cross-sections of old female TAs (two-way ANOVA, N = 10). (H) Old female TA maximum specific tetanic force production (one-tailed Student's t-test, N = 12). (I) Hang-test performance 14 days after injection of AAV-Kl or AAV-GFP, calculated relative to baseline performance (one-tailed Student's t-test, N = 15). (J) TA wet weight (as a percent of total body weight) of oldest-old female mice treated with AAV-GFP or AAV-Kl (N = 16). (K) Quantification of TA muscle-fiber CSA for oldest-old female mice (N = 12). (L) Percentage of type IIA and IIX muscle fibers in whole TA cross-sections of oldest-old female mice (N = 12). (M) Inter- and intramuscular lipid intensity in TA cross-sections of oldest-old female TAs (two-way ANOVA, N = 8). (N) Oldest-old female TA maximum specific tetanic force production (N = 15). (I) Hang-test performance 14 days after injection, calculated relative to baseline performance (N = 14). All data presented as mean ± SD (*p<0.05, **p<0.01).

The online version of this article includes the following source data and figure supplement(s) for figure 6:

**Source data 1.** Raw Data for *Figure 6D-O*, and *Figure 6—figure supplement 1A-D*.
**Figure supplement 1.** AAV-Administration in uninjured female Mice.

in the oldest-old mice (*Figure 7C*). Of note, neither circulating FGF23 protein levels, nor FGF23 transcript levels, nor the transcript levels of FGF23 and Klotho primary interactors were affected by Klotho treatment in either age group (*Figure 7—figure supplement 1*). The data suggest that the gain in Klotho may revert pathogenic FGF23 signaling through FGFR4 in old, but not oldest-old, mice (*Ho and Bergwitz, 2021*).

To better understand the differing responses to Klotho treatment between the old and the oldest-old groups, we performed Kyoto Encyclopedia of Genes and Genomes (KEGG) pathway enrichment analysis. Regulation of actin cytoskeleton, cGMP-PKG signaling pathway, and sphingolipid signaling pathway were among the top pathways in which the gene response to Klotho was opposed in the old versus oldest-old mice (*Figure 7D*, *Figure 7—figure supplement 2*, *Table 1*). A full list of affected KEGG pathways is included as *Supplementary file 1*. Significantly, the reversal of sphingosine/ceramide signaling by Klotho may have facilitated the reduction of intramyocellular lipid accumulation observed in old animals (*Figure 6C*; *Hamrick et al., 2016*). This reversal was absent in the Klotho-treated oldest-old mice, suggesting that a dysfunctional metabolism prevails with extreme aging. Further scrutiny of the top three pathways for mechanisms of Klotho's action revealed cytoskeletal regulation and calcium ion transport as sensitive to Klotho regulation in old mice (*Figure 7—figure supplement 2*). These are all tightly linked to pathogenic, non-canonical FGF23 signaling (*Xiao et al., 2019*; *Ho and Bergwitz, 2021*). In contrast, Klotho overexpression in the oldest-old mice disrupted pathways regulating cell structure, cell membrane integrity, and intercellular communication, which is consistent with the observed decreased contractile capacity (*Figure 6N*).

## Discussion

In this study, we evaluated phenotypic and gene expression trajectories across the lifespan of mice. Although declines in structure and function were observed in old mice and more prominently in oldest-old mice, alterations in the expression of genes linked to the hallmarks of aging displayed a nonlinear trajectory, with the most significant changes in gene expression profiles occurring between young and old mice. We also evaluated the ability of AAV-Kl to attenuate a sarcopenic phenotype in both old and oldest-old mice. These studies were performed only in female mice, which we found to display age-associated declines in muscle quality and function that are more comparable to clinical indices of sarcopenia when compared to male counterparts. Interestingly, we found that Klotho overexpression enhanced muscle function in the old mice. Notably, Klotho overexpression in old mice reduced pathogenic expression of genes involved with cell structure, ion homeostasis, and signaling for muscle adiposity. Unfortunately, the beneficial effects of Klotho were lost in the animals of most advanced age (*Figure 8*).

The onset of sarcopenia in mice over time and according to sex has not been well-documented (*Romanick et al., 2013*; *Argilés et al., 2015*), as studies of sarcopenia in animal models often exclude the oldest-old cohort (*Romanick et al., 2013*; *Uchitomi et al., 2019*; *van Dijk et al., 2017*). We found that decreased muscle mass and function – two classical markers of sarcopenia – become

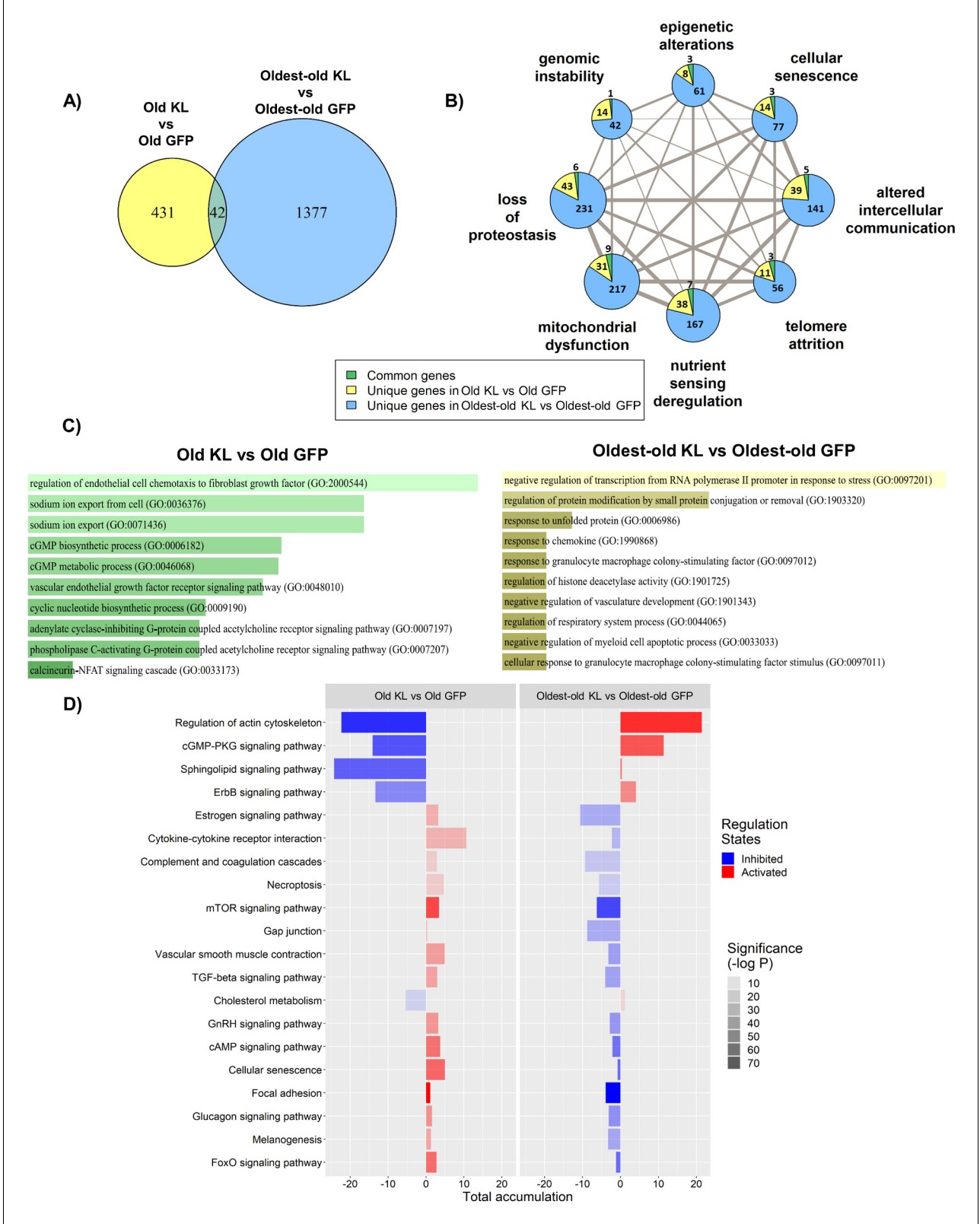

**Figure 7.** The effect of AAV-Kl administration on genes associated with hallmarks of aging is age-dependent. (**A**) Venn Diagram showing the number of differentially expressed (DE) genes between groups treated with AAV-Kl (n = 4) vs AAV-GFP (n = 4) mice. (**B**) Network plot with each node as a pie chart that denotes the total number of DE genes in that hallmark, and the wedges denote the proportion of DE genes between groups treated with AAV-Kl vs AAV-GFP for each hallmark of aging. The edge weights denote the number of genes that are common between the two connected

*Figure 7 continued on next page*

*Figure 7 continued*

hallmarks. The node sizes are proportional to the number of genes that fall into each hallmark. (C) Barplots showing GO terms associated with old vs old klotho (green), and oldest-old vs oldest-old klotho (yellow) DE genes. (D) Bar plot showing the top 20 KEGG pathways that change oppositely between old and oldest-old groups after AAV-Kl treatment ranked by largest absolute difference in total accumulation. Total accumulation is a measure of gene perturbation.

The online version of this article includes the following source data and figure supplement(s) for figure 7:

**Source data 1.** Raw Data for *Figure 7—figure supplement 1A-C*.
**Figure supplement 1.** Investigation of the Klotho-FGF23 interaction in uninjured female mice with AAV-Kl treatment.
**Figure supplement 2.** Dot plot of GO terms showing age-dependency of calcium ion transport and signaling with Klotho intervention.

prominent in the oldest-old group, whereas muscle declines in old mice are relatively modest. Additionally, the onset of sarcopenia in mice appears to be sexually dimorphic, with females displaying slightly accelerated functional declines. These findings are in agreement with one previous clinical report suggesting that sarcopenia progresses more rapidly in females (*Masanes et al., 2012*), although other studies demonstrated minimal differences between males and females (*Morley et al., 2014*; *Tay et al., 2015*). Future studies are needed to elucidate potential mechanisms by which the progression and severity of sarcopenia may be mediated by sex.

Clinical studies performed in older adults have suggested that loss of muscle function is predominantly a result of a loss of muscle quality, as opposed to a loss of muscle mass (*Goodpaster et al., 2006*). Likewise, in our study, age-related functional declines were not accompanied by a significant loss of muscle mass. Instead, muscle mass and myofiber area were, for the most part, preserved over time. Although muscle mass decreased by only approximately 10% between the old and oldest-old age groups, the accompanying decrease in peak force production was over two-fold greater.

Several transcriptional studies using functional enrichment of differentially expressed genes have reported that age-related changes play a key role in the progression of sarcopenia (*Shavlakadze et al., 2019*; *Barns et al., 2014*; *Zahn et al., 2006*). In the current study, we built on the previous body of work by using an information-based approach to describe integrative changes in gene expression over time, as defined by network entropy. Although an aging system is not necessarily closed (a now overturned criteria for an entropic system) (*Hayflick, 2007*; *Samaras, 1974*), evaluation of the interconnectedness of the PPI interactome allows for a single, integrative metric to estimate the 'disorderliness' of the system (*Menichetti et al., 2015*; *Menichetti and Remondini, 2014*). We chose to use PPI network given that the network edges are based on experimentally validated interactions. As such, the PPI network allows visualization of the functional relationship between genes, an approach that has been used in several studies that explore the mechanism of complex diseases (*Zahn et al., 2006*; *Samaras, 1974*; *Shannon, 1948*; *Teschendorff and Enver, 2017*). In contrast, gene co-expression networks contain edges that are based on correlation coefficients between gene expression values. Despite the advantages, limitations in the PPI network include the fact that mapping of genes to proteins is not necessarily one-to-one in the PPI network. Moreover, there are a number of proteins for which PPI information has not been documented.

In our study, 'disorderliness' of the gene expression profiles over time is represented by the probability of a network connection, mathematically analogous to *Shannon, 1948* entropy. Similar to a previous report using microarray gene data from human circulating T-lymphocytes, we observed that network entropy increased from young to old mice (*Menichetti et al., 2015*). Comparable to entropy change estimates in the context of stem cell differentiation, cancer, and aging (*Menichetti et al., 2015*; *Teschendorff and Enver, 2017*; *Kannan et al., 2020*; *Park et al., 2016*; *Conforte et al., 2019*), the absolute magnitude of entropy change in our study is small. Mathematically, this is not surprising given that the spatial network entropy represents the total number of possible configurations (i.e. the flexibility of the system), which has a magnitude on the order of $10^6$. The entropy of the network is dependent on the network architecture, which includes the number of nodes, the number of edges, and the edge weights (i.e. the difference in gene expression). Since fundamental features of network architecture remain relatively stable over time, the change in the magnitude of network entropy represents only a small fraction of all possible configurations (1.1%). However, this small percentage captures the change in expression of a large number of genes (~7600 genes), each of which has the potential to elicit a cascade of downstream phenotypic

**Table 1.** The top 25 GO terms associated with DE genes from old vs. old AAV-KL treated mice.

| Term | P-value | Adjusted P-value | Odds Ratio | Combined Score |
|---|---|---|---|---|
| regulation of endothelial cell chemotaxis to fibroblast growth factor (GO:2000544) | 8.17E-10 | 1.51332E-08 | 266.631818 | 5579.371 |
| sodium ion export from cell (GO:0036376) | 7.41E-15 | 2.92305E-13 | 149.472477 | 4863.183 |
| sodium ion export (GO:0071436) | 7.41E-15 | 2.92305E-13 | 149.472477 | 4863.183 |
| cGMP biosynthetic process (GO:0006182) | 1.79E-21 | 1.98668E-19 | 90.9116279 | 4343.159 |
| cGMP metabolic process (GO:0046068) | 1.41E-23 | 2.15989E-21 | 82.1943925 | 4324.53 |
| vascular endothelial growth factor receptor signaling pathway (GO:0048010) | 2.24E-39 | 2.73359E-36 | 47.4914842 | 4226.557 |
| cyclic nucleotide biosynthetic process (GO:0009190) | 6.81E-18 | 4.38219E-16 | 97.8152425 | 3866.443 |
| adenylate cyclase-inhibiting G-protein coupled acetylcholine receptor signaling pathway (GO:0007197) | 3.18E-08 | 4.17609E-07 | 221.689342 | 3827.416 |
| phospholipase C-activating G-protein coupled acetylcholine receptor signaling pathway (GO:0007207) | 3.18E-08 | 4.17609E-07 | 221.689342 | 3827.416 |
| calcineurin-NFAT signaling cascade (GO:0033173) | 8.94E-12 | 2.35047E-10 | 119.031963 | 3028.225 |
| cellular potassium ion homeostasis (GO:0030007) | 8.32E-13 | 2.71081E-11 | 100.657895 | 2799.85 |
| cellular sodium ion homeostasis (GO:0006883) | 7.48E-14 | 2.64978E-12 | 89.6743119 | 2710.338 |
| cellular monovalent inorganic cation homeostasis (GO:0030004) | 6.58E-15 | 2.68262E-13 | 82.3862069 | 2690.261 |
| cellular response to forskolin (GO:1904322) | 3.21E-09 | 5.4068E-08 | 133.309091 | 2607.271 |
| response to forskolin (GO:1904321) | 3.21E-09 | 5.4068E-08 | 133.309091 | 2607.271 |
| transmembrane receptor protein tyrosine kinase signaling pathway (GO:0007169) | 1.92E-67 | 4.69774E-64 | 16.4566607 | 2528.075 |
| ceramide metabolic process (GO:0006672) | 4.51E-24 | 7.3577E-22 | 45.96 | 2470.574 |
| regulation of cardiac conduction (GO:1903779) | 1.84E-30 | 4.49925E-28 | 35.7165215 | 2445.428 |
| peptidyl-serine dephosphorylation (GO:0070262) | 2.96E-10 | 6.14116E-09 | 103.915718 | 2279.845 |
| cyclic purine nucleotide metabolic process (GO:0052652) | 3.13E-19 | 2.38927E-17 | 51.9335548 | 2212.838 |
| calcineurin-mediated signaling (GO:0097720) | 2.63E-11 | 6.62923E-10 | 89.2694064 | 2174.732 |
| cAMP biosynthetic process (GO:0006171) | 2.28E-12 | 6.89002E-11 | 80.5221968 | 2158.455 |
| positive regulation of protein kinase B signaling (GO:0051897) | 1.97E-36 | 1.20529E-33 | 22.916681 | 1884.074 |
| regulation of myosin-light-chain-phosphatase activity (GO:0035507) | 1.09E-07 | 1.28925E-06 | 110.839002 | 1776.808 |
| cyclic nucleotide metabolic process (GO:0009187) | 1.16E-15 | 5.76736E-14 | 48.8926097 | 1681.61 |

changes. Indeed, from a physiological perspective, there is growing evidence that very small-scale mRNA changes can lead to extreme changes at the level of protein expression and cellular phenotype (*Ruzycki et al., 2015*; *Cheng et al., 2016*). In the context of cancer, for example, a single base mutation led to extensive cell remodeling that extended well beyond the predicted downstream responses, a so-called 'butterfly effect' (*Hart et al., 2015*). Others have similarly shown that cells display a sensitive dependence to initial conditions (in this case, gene expression), and that very small changes in transcriptional regulation and equilibrium may exert potent – perhaps even chaotic – impacts on disease progression (*Desi and Tay, 2019*; *Dorn, 2013*).

Whereas T-lymphocytes displayed a sharp decline in entropy in the oldest-old group, potentially suggestive of a 'survival effect' (*Menichetti et al., 2015*), we observed a plateau in entropy after old age (*Hayflick, 2007*; *Menichetti et al., 2015*; *Samaras, 1974*). The data suggest that this peak then plateau may result from a small subset of genes that display highly specific changes from young to old age, followed by a large number of small, non-specific changes between old and oldest-old ages. Our observation that network entropy in mice peaks at old age but that phenotypic changes are not prominent until later in life supports the hypothesis that phenotypic alterations are a lagging indicator of gene expression changes (*Frenk and Houseley, 2018*). Evaluation of entropy at additional time points across the lifespan would be valuable to better identify an optimal therapeutic window for sarcopenia. A therapeutic challenge lies in how to promote healthier aging and best

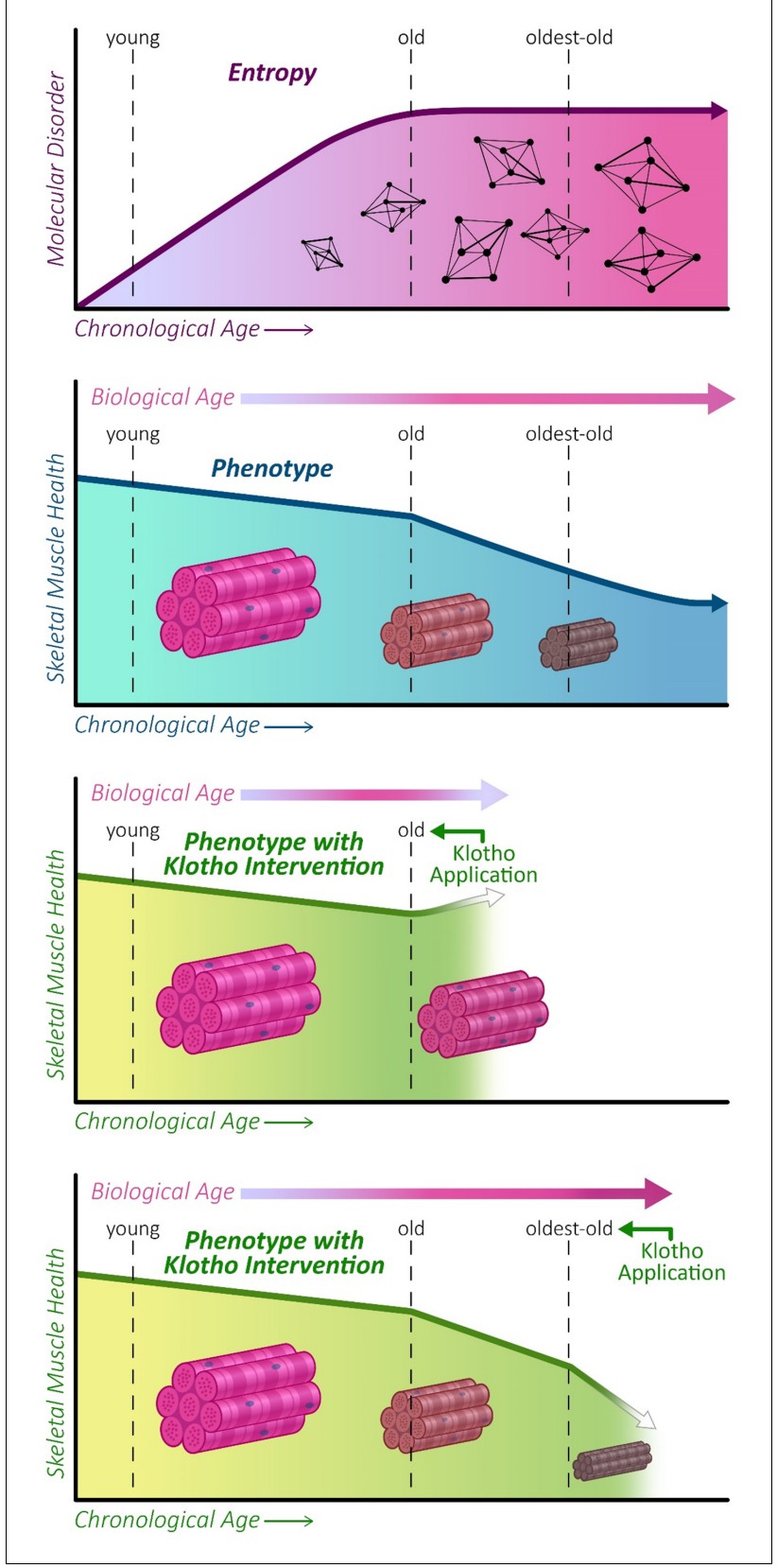

**Figure 8.** Graphical abstract. Entropy increases with increasing age before plateauing at old age (21–24 months) in mice (top panel). This is concomitant with a decline in skeletal muscle function, which continues to progress into

*Figure 8 continued on next page*

*Figure 8 continued*

oldest-old age (second panel). We show that AAV-Kl administration can rescue muscle functional declines when administered to old mice (third panel), but this effect is lost when AAV-Klotho is delivered to oldest-old (27–29 months) mice (bottom panel).

---

address the root of these gene expression changes before a sarcopenic phenotype develops. Moreover, although here we focused on quantification of entropy at the gene-level, given the progressive loss of histological and functional muscle integrity, it would be interesting to determine whether entropy at the proteomic level (e.g. using mass spectrometry) is able to capture entropic changes even into very old age.

Given the aforementioned observations, we tested whether modulation of genes associated with the hallmarks of aging is capable of restoring a more youthful skeletal muscle profile. For this, we chose systemic overexpression of Klotho via AAV. AAV allows for consistent modulation of Klotho levels without the need for frequent re-administration and without the challenges of recombinant Klotho protein administration (*Cheikhi et al., 2019*). Advanced clinical trials using a similar capsid construct are underway (*George, 2017*; *Kishnani et al., 2019*), demonstrating the potential translatability of this method. We found that AAV-Kl administration modulated all hallmarks of aging in old

**Table 2.** The top 25 GO terms associated with DE genes from old vs. old AAV-Kl treated mice.

| Term | P-value | Adjusted P-value | Odds Ratio | Combined Score |
|---|---|---|---|---|
| negative regulation of transcription from RNA polymerase II promoter in response to stress (GO:0097201) | 0.000331 | 0.09424241 | 13.1472045 | 105.3348 |
| regulation of protein modification by small protein conjugation or removal (GO:1903320) | 0.00023 | 0.07658759 | 9.8657932 | 82.66108 |
| response to unfolded protein (GO:0006986) | 5.3E-06 | 0.00896227 | 5.72185053 | 69.50717 |
| response to chemokine (GO:1990868) | 0.006044 | 0.307829996 | 13.1300353 | 67.07703 |
| response to granulocyte macrophage colony-stimulating factor (GO:0097012) | 0.006044 | 0.307829996 | 13.1300353 | 67.07703 |
| regulation of histone deacetylase activity (GO:1901725) | 0.006044 | 0.307829996 | 13.1300353 | 67.07703 |
| negative regulation of vasculature development (GO:1901343) | 0.006044 | 0.307829996 | 13.1300353 | 67.07703 |
| regulation of respiratory system process (GO:0044065) | 0.006044 | 0.307829996 | 13.1300353 | 67.07703 |
| negative regulation of myeloid cell apoptotic process (GO:0033033) | 0.006044 | 0.307829996 | 13.1300353 | 67.07703 |
| cellular response to granulocyte macrophage colony-stimulating factor stimulus (GO:0097011) | 0.006044 | 0.307829996 | 13.1300353 | 67.07703 |
| regulation of microtubule nucleation (GO:0010968) | 0.006044 | 0.307829996 | 13.1300353 | 67.07703 |
| cellular response to chemokine (GO:1990869) | 0.006044 | 0.307829996 | 13.1300353 | 67.07703 |
| RNA stabilization (GO:0043489) | 0.000787 | 0.169749244 | 7.17396343 | 51.27482 |
| chaperone-mediated protein complex assembly (GO:0051131) | 0.000787 | 0.169749244 | 7.17396343 | 51.27482 |
| mitochondrial translation (GO:0032543) | 1.36E-06 | 0.004970703 | 3.63077317 | 49.05655 |
| 3'-UTR-mediated mRNA destabilization (GO:0061158) | 0.003735 | 0.29111327 | 8.75813296 | 48.95746 |
| primary miRNA processing (GO:0031053) | 0.003735 | 0.29111327 | 8.75813296 | 48.95746 |
| cellular response to leucine starvation (GO:1990253) | 0.003735 | 0.29111327 | 8.75813296 | 48.95746 |
| endothelial tube morphogenesis (GO:0061154) | 0.003735 | 0.29111327 | 8.75813296 | 48.95746 |
| negative regulation of inclusion body assembly (GO:0090084) | 0.003735 | 0.29111327 | 8.75813296 | 48.95746 |
| mitochondrial gene expression (GO:0140053) | 4.92E-05 | 0.025776303 | 4.79736196 | 47.58752 |
| regulation of transcription from RNA polymerase II promoter involved in heart development (GO:1901213) | 0.010022 | 0.41209955 | 9.84699647 | 45.32527 |
| mitochondrial electron transport, ubiquinol to cytochrome c (GO:0006122) | 0.002072 | 0.281362729 | 7.30242982 | 45.12396 |
| mitochondrial translational elongation (GO:0070125) | 8.63E-06 | 0.00896227 | 3.6976622 | 43.11427 |
| mitochondrial translational termination (GO:0070126) | 1.22E-05 | 0.00896227 | 3.59162667 | 40.63 |

and oldest-old animals, although the magnitude of change was manifold times greater in the oldest-old mice. This enhanced response in oldest-old mice was not, however, accompanied by a greater therapeutic benefit. Instead, though we observed a significant increase in muscle function in old mice receiving AAV-Kl, intervention failed to produce a benefit in the oldest-old mice.

Observations of treatment resistance have been reported in advanced age and critically ill populations in other pharmaceutical treatment studies (*Rennie, 2009*; *Burd et al., 2013*). In humans, dieting and exercise intervention studies reveal blunted muscle protein synthesis in an older population (greater than 60 years old) (*Breen and Phillips, 2011*; *Durham et al., 2010*). Interestingly, this coincides with the same stage in which we observed a tipping point in network entropy (*Figure 3—figure supplement 4C*). The lack of beneficial response to treatment with increased age is hypothesized to stem from deficiencies in downstream anabolic pathways, that is 'anabolic resistance' (*Burd et al., 2013*; *Glover et al., 2008*; *Burd et al., 2012*). In order to identify potentially compromised signaling pathways in oldest-old mice that may preclude a therapeutic benefit of AAV-Kl administration, we compared KEGG pathway activation differences across treatment groups. Old and oldest-old mice displayed opposing responses to AAV-Kl in pathways associated with cellular structure, lipid membrane integrity, and intercellular communication. When we further probed age-dependent pathways that were responsive to Klotho treatment, we found that cytoskeletal signaling, calcium ion transport (cGMP-PKG signaling), and signaling for muscle adiposity (sphingolipid signaling) displayed biphasic effects (*Figure 7D*). Overwhelmingly, the top GO terms for genes that were decreased by Klotho therapy in the old mice are potentially affected by aberrant FGF signaling, including genes associated with calcineurin (*Figure 7C*, *Table 2*). Soluble Klotho binds FGF23 to increase its affinity for FGFR1 and prevents its interaction with FGFR4 to reduce expression of the pathogenic calcineurin/NFAT transcriptome (*Ho and Bergwitz, 2021*; *Han et al., 2020*). Indeed, FGF23/FGFR4 signaling is associated with muscle pathogenesis in the heart as well as renal fibrosis (*Han et al., 2020*; *Hao et al., 2021*), processes that, if active in skeletal muscle, could explain the decline in muscle quality over time. Our findings support the hypothesis that restoration of normal calcium signaling and calcineurin/NFAT interactome may be important in the development of therapeutics for sarcopenia. Although Klotho is well-known to regulate these pathways by promoting interactions with FGF23 and FGFR1 (70, 71), the age-dependency of Klotho treatment has not been shown.

We also found that the sphingolipid signaling pathway was inhibited in old, but not in the oldest-old, mice following Klotho treatment. Previous aging studies in *C. elegans* and *D. melanogaster*,

**Table 3.** Top 25 GO terms associated with DE genes from oldest-old vs. oldest-old AAV-Kl-treated mice.

| | Klotho mediated Bi-phasic KEGG Pathways Gene Expression | | | | | | | |
| --- | --- | --- | --- | --- | --- | --- | --- | --- |
| | Old + log2FC | | Oldest-Old + log2FC | | | | | |
| Regulation of actin cytoskeleton | F2 | -1.92627 | Bcarl | 0.40772 | Rras | 0.30578 | Rhoa | 0.08657 |
| | Kng2 | -1.09411 | Arpc3 | 0.22899 | Pik3rl | -0.94428 | Itga5 | 0.30051 |
| | Diaph2 | 0.25901 | Brkl | 0.17587 | Rafl | -0.19865 | Hras | 0.22869 |
| | | | Itga9 | 0.39371 | Arhgap35 | -0.24801 | Pfn1 | 0.19127 |
| | | | Pdgfrb | -0.35413 | Pip4k2c | -0.16982 | Pfn2 | -0.20196 |
| | | | Itgax | 1.6131 | Apc2 | -0.52877 | | |
| | | | Arpc2 | 0.1243 | Cdc42 | 0.11714 | | |
| cGMP-PKG signaling | F2 | -1.9263 | Dgke | -0.3491 | Pik3rl | -0.94428 | Cyth1 | 0.13784 |
| | Lpar3 | 1.82228 | Pdgfrb | -0.35413 | Rafl | -0.19865 | Rhoa | 0.08657 |
| | | | Adcy9 | -0.20643 | Dnm3 | -0.54357 | Hras | 0.22869 |
| | | | Rras | 0.30578 | Pla224e | -0.25837 | | |
| Sphingolipid signaling pathway | Kng2 | -1.0941 | Sgppl | -0.1274 | Pik3rl | -0.94428 | Hras | 0.22869 |
| | | | Map3k5 | 0.27371 | Rafl | -0.19865 | | |
| | | | Ctsd | 0.17911 | Rhoa | 0.08657 | | |

suggest that inhibition of sphingolipid synthesis may be beneficial for maintaining proper lipid homeostasis and even extending lifespan (*Hla and Dannenberg, 2012*; *Johnson and Stolzing, 2019*). Alterations in the sphingolipid signaling pathway could explain our observation of decreased lipid accumulation in old muscle following Klotho treatment, but a reversed trend in oldest-old mice. Further investigation to more deeply probe this possible mechanism is warranted. It is also possible that the increased lipid accumulation with Klotho treatment in oldest-old mice may be due to a dysregulation in mitochondrial metabolism (*Johannsen et al., 2012*). Indeed, many of the DE genes in oldest-old Klotho-treated mice were related to mitochondrial translation and electron-transport pathways (*Table 3*). Metabolic dysregulation would also be consistent with the trend of intramuscular lipid accumulation, as disrupted carbohydrate metabolism could cause an intracellular buildup of stored lipids (*Johannsen et al., 2012*). Additional mechanistic studies that link FGFR4 activity to sphingolipid/ceramide promoted muscle adiposity (*Hamrick et al., 2016*) and increased expression of calcium and potassium channels are warranted, as genes associated with these pathways were also the most greatly suppressed by AAV-Kl in an age-dependent manner.

Taken together, the data presented here suggest that intervention with AAV-Kl may be more effective in slowing the progression of sarcopenia at an earlier timepoint, rather than rescuing advanced pathology, at which time the transcriptomic response to intervention appears to be more stochastic. An interesting area of future investigation includes the determination of whether network entropy and PPI network architecture may be predictive of the efficacy of therapies designed to counteract the effect of time on skeletal muscle health and function. As an extension of this work, it would also be interesting in future studies to determine whether upregulation of Klotho at a younger age could attenuate functional declines into old, and possibly even oldest-old, age.

# Materials and methods

## Key resources table

| Reagent type (species) or resource | Designation | Source or reference | Identifiers | Additional information |
|---|---|---|---|---|
| Gene (*Mus musculus*) | *klotho* (*Kl*) | MGI | MGI:1101771 | |
| Strain, strain background (mouse) | C57BL/6J | NIA | RRID:IMSR_JAX:000664 | |
| Cell line (*Homo sapiens*) | HEK-293H | Thermo Fisher | Cat#: 11631017 RRID:CVCL_6643 | For vector construction |
| Antibody | Anti-Laminin (Rabbit polyclonal) | Abcam | Cat#: ab11575, RRID:AB_298179 | IF(1:500) |
| Antibody | Anti-Type IIA Fibers (mouse monoclonal) | DSHB | Cat#: SC-71, RRID:AB_2147165 | IF(1:100) |
| Antibody | Anti-Type IIB Fibers (mouse monoclonal) | DSHB | Cat#: BF-F3, RRID:AB_2266724 | IF(1:100) |
| Antibody | Anti-Collagen IV (rabbit polyclonal) | Abcam | Cat#: ab6586, RRID:AB_305584 | IF(1:500) |
| Antibody | Anti-Klotho capture antibody (goat polyclonal) | R and D Systems | Cat#: AF1819, RRID:AB_2296612 | MSD-ELISA(4 µg/mL) |
| Antibody | Anti-Klotho detection antibody (goat polyclonal) | R and D Systems | Cat#: BAF1819, RRID:AB_2131927 | MSD-ELISA(1 µg/mL) |
| Antibody | Anti-CD31 (rat monoclonal) | Thermo Fisher | Cat#:1 11–0311081, RRID:AB_465011 | FACS(1:500) |
| Antibody | Anti-CD45 (mouse monoclonal) | Thermo Fisher | Cat#: 11-0451-81, RRID:AB_465049 | FACS(1:500) |
| Antibody | Anti-Sca1 (rat monoclonal) | Thermo Fisher | Cat#: 25-5981-82, RRID:AB_469669 | FACS(1:33) |

*Continued on next page*

*Continued*

| Reagent type (species) or resource | Designation | Source or reference | Identifiers | Additional information |
|---|---|---|---|---|
| Antibody | Anti-α−7 (rat monoclonal) | Thermo Fisher | Cat#: MA5-23555, RRID:AB_2607368 | FACS(1:200) |
| Peptide, recombinant protein | Klotho | R and D Systems | Cat#: 1819 KL-050 | MSD-ELISA |
| Other | Lipidtox stain | Thermo Fisher | Cat#: H34476 | IF(1:500) |
| Other | DAPI stain | Invitrogen | Cat#: D1306 RRID:AB_2629482 | IF(1:1000) |
| Other | Trichrome Stain Solution | Sigma | Cat#: HT10516 | |
| Other | Weigarts's Haemotoxylin | Poly Scientific R and D | Cat#: S216B | |
| Other | Bouin's solution | Sigma | Cat#: HT101128 | |
| Commercial assay or kit | Mouse Klotho ELISA | Cloud-Clone Corp. | Cat#: SEH757Mu | Lot: L180223640 |
| Commercial assay or kit | FGF23 ELISA | Abcam | Cat#: ab213863 | Lot: GR3326863 |
| Commercial assay or kit | AllPrep DNA/RNA 96 kit | Qiagen | Cat#: 80311 | |
| Chemical compound, drug | Poly/Bed 812 | Polysciences | Cat#: 08792–1 | |
| Chemical compound, drug | Cardiotoxin | Sigma | Cat#: 217503 | |
| Recombinant DNA reagent | AAV-GFP | *Strobel et al., 2019* | | AAV8-LP1-eGFP |
| Recombinant DNA reagent | AAV-Klotho | This paper | | AAV8-LP1-mKlotho Mouse Klotho version of AAV vector |
| Sequence-based reagent | LP-1 promoter (forward) | This paper | PCR primers | GACCCCC TAAAA TGGGCAAA |
| Sequence-based reagent | LP-1 promoter (reverse) | This paper | PCR primers | TGCCCCAGC TCCAAGGT |
| Biological sample (*M. musculus*) | Mouse gastrocnemius muscle | NIA | | Freshly dissected from C57BL/6J mice |
| Biological sample (*M. musculus*) | Mouse serum | NIA | | Freshly dissected from C57BL/6J mice |
| Biological sample (*M. musculus*) | Mouse tibialis anterior muscle | NIA | | Collected fresh from C57BL/6J via cardiac puncture |
| Software, algorithm | STAR_2.7.0a | *Dobin et al., 2013* | RRID:SCR_004463 | |
| Software, algorithm | R | R Project for Statistical Computing | RRID:SCR_001905 | |
| Software, algorithm | ggplot2 | R Project for Statistical Computing | RRID:SCR_014601 | R package |
| Software, algorithm | clusterProfiler | Bioconductor | RRID:SCR_016884 | R package |
| Software, algorithm | VennDiagram | R Project for Statistical Computing | RRID:SCR_002414 | R package |

*Continued on next page*

*Continued*

| Reagent type (species) or resource | Designation | Source or reference | Identifiers | Additional information |
|---|---|---|---|---|
| Software, algorithm | igraph | R Project for Statistical Computing | RRID:SCR_019225 | R package |
| Software, algorithm | BioMart | BioMart Project | RRID:SCR_002987 | R package |
| Software, algorithm | Fiji-ImageJ | NIH | RRID:SCR_002285 | |
| Software, algorithm | MuscleJ Macro | *Mayeuf-Louchart et al., 2018* | RRID:SCR_020995 | |
| Software, algorithm | GraphPad Prism v9.0 | GraphPad | RRID:SCR_002798 | |
| Software, algorithm | R ShinyApp resource | This paper | https://sruthisivakumar.shinyapps.io/HallmarksAgingGenes/ | Gene classification based on hallmarks of aging. Building details are provided in 'Hallmarks of aging genes classification' (Materials and methods). |
| Software, algorithm | Python Streamlit | This paper | https://network-entropy-calculator.herokuapp.com/ | Network entropy calculator app hosted on Heroku |
| Software, algorithm | GitHub | This paper | https://github.com/sruthi-hub/sarcopenia-network-entropy | Code for RNA - seq network entropy provided with explanation |
| Other | STRING | STRING Consortium | RRID:SCR_005223 | Protein interaction database |

## Animals and ethics

All animal experiments were approved by the University of Pittsburgh's Institutional Animal Care and Use Committee. These experiments were conducted in accordance with protocol 17080802 (University of Pittsburgh ARO: IS00017744). Experiments were performed using young (3–6 months), middle-aged (10–14 months), old (21–24 months), and oldest-old (27–29 months) male and female C57 BL/6J mice. Mice were obtained from the NIA Rodent Colony, Jackson Laboratories, and Charles River Laboratories.

## Study rigor

All mice were housed in pathogen-free conditions under a 12 hr light and 12 hr dark cycle. Animals in the AAV portion of the study were housed and experimented on under stricter biosafety regulations (Biosafety level II). Animals with obvious health problems (i.e, tumors, malocclusion, etc.) were eliminated prior to inclusion into the study when possible and were excluded from endpoint analyses if pathologies were noted during experimentation. Given the large amount of variability in physical endurance capacity, animals were randomized into cohorts only if they met the criteria of falling into the 25th-75th percentile for the four-limb hang test (described below). This allowed for elimination of outliers that displayed baseline physical performance falling in the first and fourth quartiles, which may confound results. All animals meeting criteria for inclusion were then sorted by restricted randomization, ensuring comparable baseline physical performance across treatment groups. For in vivo experiments, each experiment was repeated across a minimum of two cohorts, and

experimenters performing endpoint analysis were blinded to the experimental groups. Power analyses were performed a priori when possible, assuming a two-sided alternative hypothesis with alpha = 0.05% and 80% power (G*Power 3.1.9.2). Based on the prior findings from the laboratory on peak tetanic force in young and aged animals and after adjusting for unforeseen circumstances estimated at 20%, the final sample size was calculated to be 6–10 animals/group. Each endpoint presents data for biological replicates (individual animals). Some datapoints represent mean values calculated from technical replicates – multiple images captured from one animal sample. This is mentioned in the methods subsections below. The raw data for all experiments reported are included in the accompanying Source Data file.

## Cryopreservation and preparation of skeletal muscle tissue

Post euthanasia , the tibialis anterior (TA) muscles of mice were carefully extracted and wet weight was measured using a standard balance. Immediately after weighing, TAs were frozen in liquid nitrogen-cooled 2-methyl butane for one minute, and stored at −80°C. Slides for histological analyses were prepared using a Thermo Fisher CryoStarNX50 cryostat. Tissue was cut at 10 µm with the cryostat set at −20°C. Each slide captured equidistant tissue sections throughout the entire length of the TA. Slides were stored in a −80°C freezer until use.

## Immunofluorescence staining and imaging

One slide from each experimental replicate was thawed and immediately fixed by covering sections with 2% paraformaldehyde using a pipette for 10 min. After fixing, slides were washed with 1X-Phosphate Buffered Saline (PBS) three times for two minutes each. A hydrophobic barrier was then drawn around the sections, and they were permeabilized with 0.1% triton-X (Fluka 93420) in PBS for 15 min, followed by a 1-hr blocking step using 0.1% triton-X plus 3% Bovine Serum Albumin (BSA, Sigma A7906) in PBS. Primary antibodies were diluted (*Table 4*) in a solution of 0.1% triton-X, 3% BSA, and 5% normal goat serum in PBS then added onto the sections and incubated overnight at 4°C. One negative control slide per staining set was generated by deleting the primary antibody in the antibody solution. The secondary antibodies used were goat anti-rabbit and goat anti-rat IgG cross-adsorbed antibodies (Invitrogen) and were diluted in the same base solution as the primary antibodies. After washing three times for 2 min in PBS, the secondary antibodies were added to the sections for 1 hr and incubated in the dark at room temperature for one hour. The slides were washed with PBS three times and DAPI stain (Invitrogen D1306, 1:500 dilution in PBS) was added onto the slides for 2 min. Next, slides were washed twice with PBS, dried, and mounted with coverslips using Gelvatol mounting medium (Source: Center of Biologic Imaging, University of Pittsburgh). Slides were allowed to dry for at least 24 hr at 4°C prior to imaging.

Slides were imaged using a Zeiss Observer Z1 semi-confocal microscope. For uninjured slides, regions of interest were randomly selected by navigating to an imaging site of normal nuclear density based on the DAPI channel and capturing an image. At least three images (technical replicates) were captured per slide (each slide contained tissues from one animal), and images were collected at ×20 magnification. Negative control slides were used to threshold for the signal intensity and to set the exposure time for individual channels. The exposure time was kept consistent for each set of stains based on the negative control. For injured tissues, the injury site was located by finding the

**Table 4.** Antibody/stain list.
Primary antibodies and corresponding dilutions used for immunofluorescence imaging.

| Antibody/stain | Source | Dilution |
| --- | --- | --- |
| Rabbit anti-Laminin | Abcam ab11575 | 1:500 |
| Lipidtox Red Stain | Invitrogen H34476 | 1:200 |
| Collagen IV | Abcam ab6586 | 1:500 |
| Type IIA Muscle Fibers | DSHB SC-71 | 1:100 |
| Type IIB Muscle Fibers | DSHB BF-F3 | 1:100 |
| DAPI Stain | Invitrogen D1306 | 1:1000 |

localized areas of highest nuclear infiltration. Images were then collected at the injury site and at least three images were collected per slide. Intensity was analyzed for Collagen IV using the Zen software and was normalized to image area.

## Immunofluorescence staining, imaging, and analysis for muscle fiber types

Fiber-type staining method was adapted from a previously published protocol (*Rao and Mohanty, 2019*). Briefly, frozen muscle sections were allowed to dry completely at room temperature, then a blocking solution of 0.5% triton-x, 1% BSA, and 10% goat serum in PBS was added and slides were incubated for 1 hr at room temperature. Antibodies for laminin, type IIA fibers, and type IIB fibers were prepared in 0.1% Tween-20, 1% BSA, and 5% goat serum according to the dilutions in *Table 4*. This antibody mixture was added onto the sections and incubated overnight at room temperature. Next, slides were washed three times in PBS for five minutes each, then fixed in 10% neutral buffered formalin for 10 min, followed by three more washes. Corresponding secondary antibodies (goat anti-mouse IgM and IgG1, goat anti-rabbit, Invitrogen) were prepared at 1:500 dilution in the same solution used for primary antibodies. Sections were incubated for 1 hr and washed three more times. Then, nuclei were stained with DAPI as above followed by washing, airdrying, and coverslipping using Gelvatol mounting medium.

Whole muscle section images were captured at 20X using a Zeiss Axio Observer with tile stitching settings. One image was captured per animal. These images were analyzed using the Fiji-ImageJ MuscleJ macro developed by Danckaert and Mayeuf-Louchart (*Mayeuf-Louchart et al., 2018*) and downloaded from Github, which gave outputs for fiber number, area, and type.

## Intramuscular vs. intermuscular adipose tissue quantification

Inter- and intra-muscular lipid deposition was quantified using Fiji-ImageJ software. The laminin rings were converted to a binary scale with positive particles indicating intra-muscular regions. Regions of Interest (ROI) were set and converted to a mask as selected by the positive particles. Total lipid deposition was calculated by measuring the total lipid intensity and then normalizing it to the image area. Intra-muscular lipid deposition was calculated by overlaying the ROI mask onto the lipidtox image where lipid intensity was measured specific to the locations marked by the ROI prior to normalization. Inter-muscular lipid deposition was determined using the resulting difference between total lipid deposition and intra-muscular lipid deposition. Mean values were calculated from three images captured for each sample.

## Masson's trichrome staining and analysis

Staining was conducted using glass Coplin jars. Slides containing frozen tissue sections were fixed in acetone for 10 min followed by two 5-min PBS washes. The slides were then dipped in 95% ethanol and air-dried for 1 hr. Refixing was done in Bouin's solution (Sigma HT101128) for 15 min at 56 °C, then the slides were rinsed under running tapwater for 1–5 min until excess yellow coloring dissipated. Next, Wiegert's Hematoxylin working solution (Poly Scientific S216B) was added for 5 min followed by another tapwater rinse. Then, the slides were stained with Trichome Stain Solution (Sigma HT10516) for 5 min and rinsed with tapwater. Finally, the slides were dehydrated quickly through 95% ethyl alcohol, 100% alcohol (twice), cleared with Xylenes, and mounted with Clearvue Mountant (ThermoScientific 4211). Images were captured with brightfield at 10x using a Nikon Eclipse 50*i*.

Fiji-ImageJ was used for analysis. First, a threshold was set to remove blank background. The collagen signal was isolated by using color deconvolution and thresholding. The area of the collagen signal was then compared to the net area of the muscle section in the image. Mean collagen coverage was calculated for each sample from three separate images.

## Transmission electron microscopy

Tibialis anterior muscles from the experimental groups were fixed in 2.5% Glutaraldehyde for 24 hr in 4°C. The samples were then washed with PBS three times and cut longitudinally into small pieces. The tissue pieces were submerged in post-fixation aqueous solution comprising of 1% osmium tetroxide, 1% $Fe_6CN_3$ for 1 hr, following which they were washed in PBS three times. The tissues were then treated with aqueous 1% osmium tetroxide, 1% $Fe_6CN_3$ for 1 hr. Samples were then

washed with PBS three times and then dehydrated with a series of 30–100% ethanol. Following this, the samples were embedded in by inverting Poly/Bed 812 (Polysciences, Warrington, PA) embedding resin-filled BEEM capsules on top of the samples. The resin blocked were then cured twice— once overnight at 37°C, and next at 65°C for 2 days. After the curing procedure, the samples were peeled from the coverslip and cross-sectioned ultrathin on a Reichart Ultracut E microtome. Samples were submerged in uranyl acetate for 10 min and 1% lead citrate for 7 min. The investigators performing imaging were blinded to the hypotheses. Data analysis on mitochondrial ultrastructure was performed by a blinded investigator as well following a previously established classification method (*Owen et al., 2019*).

## Four-limb hang test

Whole body endurance was measured using a modified version of a previously established protocol (*Carlson, 2011*). The four-limb hang test was used to measure muscular endurance non-invasively and, thus, characterize performance before and/or after treatment. Mice were suspended upside down from a steel mesh grid (1 cm x 1 cm squares) above a custom-built chamber (~30 cm high) with appropriate padding to prevent harm. Four limb strength was evaluated using a Hang Impulse (HI) score (*bodyweight in grams*time spent hanging*). The time spent hanging was recorded from the time of inversion until all paws are released from the grid. Prior to recorded trials, mice were acclimatized to the grid and the chamber for 5 min. Mice were subjected to five trials in one session with a rest of at least 5 min between trials. Sessions were conducted at roughly the same time of the day (between 10 am and 12 pm) and in the same progression across trials (i.e. the first mouse on session one was also the first on session two). Evaluation of performance was calculated as the average HI of three best trials. Baseline endurance capacity was evaluated prior to experimentation.

For muscle regeneration experiments, injured animals treated with AAV-GFP or AAV-Kl, hang tests were performed before and after AAV- injections, immediately prior to injury, and then again at 1, 7, and 14 days post-injury. For uninjured animals, hang tests were conducted prior to AAV injection, then again at 1, 7, and 14 days after injection.

## In Vivo contractile testing

TA muscle strength of animals in the aged cohorts and treatment cohorts was evaluated using an in vivo testing apparatus (Model 809B, Aurora Scientific, Aurora, ON, Canada). Anesthetized mice underwent surgery to expose the peroneal nerve for stimulation and the Achilles tendon was severed to prevent a counteracting response. The foot was secured onto a footplate in 20° of plantar flexion for maximal responses (*Distefano et al., 2013*). Initially, single twitch stimulation tests were conducted to measure the peak twitch, time to peak twitch, and the half-relaxation time. Shortly after, tetanic stimulations (350 ms train) at 10, 30, 50, 80, 100, 120, 150, 180, and 200 Hz with 2-min intervals between contractions were induced to obtain the force-frequency curves. Post tetanic stimulation, animals were given a recovery period of 10 min before proceeding to the high-frequency fatiguing protocol wherein muscles were activated every four seconds by a stimulus train at 100 Hz for 7 min. After completion, force recovery was evaluated at 5 and 10 min timepoints. Results of both the single and tetanic stimulations were collected in torque (mN-m) and the absolute force values (mN) were extrapolated by dividing by the length of the footplate (0.03 m). Values were then normalized to physiological muscle cross-sectional area (CSA measured in mm$^2$), as described by *Brooks and Faulkner, 1988*. Physiological CSA was calculated as $\frac{muscle\ mass}{muscle\ density * muscle\ length}$ (*Sacks and Roy, 1982*), with muscle density approximated as 1.06 g/cm$^3$ as per previous reports (*Méndez, 1960*). Results of the high-frequency fatigue and recovery test were analyzed as percentage values of the initial response.

## Hallmarks of aging genes classification

First, we compiled a list of key functions and changes for all systemic hallmarks of aging and cellular hallmarks of aging (*López-Otín et al., 2013*; *Frenk and Houseley, 2018*; *Rebelo-Marques et al., 2018*; *DiLoreto and Murphy, 2015*). Next, we created multiple gene lists by searching for different combinations of gene ontology (GO) terms on the MGI GO search database (*Supplementary file 1*). Batch query forms were used to identify genes for each hallmark. A table of (genes) x (hallmarks)

was formed by genes classified into different hallmarks of aging. Note that GO terms associated with multiple hallmarks of aging were accounted for in this process.

From this step onwards, we focused on each hallmark of aging, as opposed to a specific gene or a GO term. The goal of the study was to identify and target the genes associated with the hallmarks of aging with or without intervention. We created an R ShinyApp webpage for the research community to classify genes into corresponding hallmarks of aging and vice versa (https://sruthisivakumar.shinyapps.io/HallmarksAgingGenes/).

## RNA-seq analysis

We performed RNA sequencing on gastrocnemius muscle collected from female mice injected with AAV-GFP and AAV-Kl. Four animals were used for each of the five groups: young GFP, old GFP, old Klotho, oldest-old GFP, and oldest-old Klotho. Raw data has been submitted to GEO database (GSE156343). Raw fastq files were converted to gene counts after alignment using STAR_2.7.0a to reference genome GRCm38/mm10 (chromFa.fa, UCSC). The visualizations were generated using R packages, ggplot2, clusterProfiler, VennDiagram, and igraph. DE analysis was performed with Deseq2. DE genes with p-value less than 0.05 were defined as significantly different.

To identify genes associated with the hallmarks of aging, a list of 77,708 genes derived from Mouse Genome Informatics' batch query was generated based on the GO terms related to cell biology of aging. The ratio of DE genes that changed under each hallmark was visualized in the barplot and then network plot. Shannon network entropy was computed after mapping genes to proteins using Biomart in R.

## Network entropy methodology and interpretation

### Pre-processing RNA-seq to protein-protein interaction (PPI) networks

Raw gene count matrix was used to generate PPI networks. For each sample, we generated a node-list and an edge-list. Absolute gene expression values were used to create the node-list, while the edge-list was generated with mouse PPI networks obtained from STRING database version 11.0. If the gene counts were zero, then the nodes were not considered part of the PPI network. This process was done for all the genes, as well as for only the genes classified into hallmarks of aging.

### Network entropy interpretation

In the field of information theory, Shannon entropy is the measure of surprise of an outcome. For example, a fair coin toss has two equally probable outcomes. This would be the case of maximum surprise, or maximum Shannon entropy. On the other hand, if the coin is rigged, and probability of obtaining heads is 0.9 then one will not be as surprised to get heads. This is the case of lower Shannon entropy. When this concept is extrapolated to graph theory, we aimed to predict the presence or absence of an edge in the graph. If network entropy is high, this means that it is harder to predict the network connections with the knowledge one has. A higher network entropy is bound to happen when the knowledge one has about the system is insufficient or highly variable.

The principle of maximum entropy states that the probability distribution best representing the current state of knowledge on a system is the one that maximizes the Shannon entropy, given the prior knowledge about the system (*Cimini et al., 2019*). Using this method, we gain maximally unbiased information in the absence of complete knowledge. In the content of network biology, the ensemble of most probable network configurations are the ones that have maximum entropy, given certain constraints.

Our goal was to capture the change in gene expression distribution across age groups using a network entropy estimate, as has been previously described (*Menichetti et al., 2015*; *Menichetti and Remondini, 2014*). We obtained nodelists from the RNA-seq experiment, and edgelists obtained from existing databases (e.g. STRING) that have PPI network. In addition, we accounted for relative gene expressions by weighting the edges of the network to incorporate the current knowledge we have about the PPI network connections. Hence, we hypothesized that changes in gene expressions will affect the number of nodes, number of edges and the edge weights–a phenomenon that could potentially be captured by the network entropy metric.

Conventionally, maximum entropy is used to predict the probability distribution. But in this study, we used network entropy values corresponding to the most probable ensemble of PPI networks.

This allowed us to quantify the change in transcriptional noise and variation in gene expressions into a single integrative metric.

## Network entropy calculation

Network entropy is a constraint-based optimization problem that is solved using Lagrangian multipliers. The objective function to maximize is the Shannon entropy of the network. A value of network entropy will correspond to a probability distribution for edges. There are two constraints for the optimization problem. The first constraint imposes a degree sequence of the network, whereas the second constraint imposes restriction on the edge weights of the network. The concept is explained with an arbitrary simple example, as shown below.

| Gene counts for node-list (15 genes) | | | | | | | | | | | | | | | Age and # nodes (n) |
|---|---|---|---|---|---|---|---|---|---|---|---|---|---|---|---|
| **500** | **700** | **0** | **0** | **0** | **0** | **0** | **100** | **58** | **350** | **0** | **675** | **10** | **509** | **0** | **Young (n = 8)** |
| 502 | 704 | 0 | 1 | 2 | 3 | 0 | 95 | 59 | 375 | 0 | 450 | 0 | 458 | 0 | Young (n = 10) |
| 600 | 690 | 0 | 1 | 0 | 5 | 0 | 110 | 65 | 378 | 2 | 567 | 9 | 450 | 0 | Old (n = 11) |
| 595 | 721 | 0 | 0 | 0 | 9 | 0 | 100 | 0 | 300 | 0 | 604 | 0 | 425 | 0 | Old (n = 7) |

Let edge-list be: (1-3), (3-7), (5-8), (8-9), (9-12), (10-14).

### Constraint 1

The degree sequence was fixed based on the edge-lists. Zeroes are not considered in the network as nodes.

### Constraint 2

The edge weights were calculated by taking the difference between gene counts between the connected nodes. To normalize for different number of nodes in the network, the edge weights are binned with number of bins equal to square root of the number of nodes in the network.

In the arbitrary example above, there would be sqrt(n)~3 bins. [0–240, 241–482, 483–721]. The binning distribution will vary according to the relative gene expression changes. If the binning is similar for all the samples, then network entropy values will be very close to each other. If the binning is different, network entropy values will be different.

## Network characterization

The main global features are described for both *Figure 3E* and *Figure 3—figure supplement 2* in *Supplementary file 2*. These global features include: number of nodes, number of edges, spatial entropy, link density, the number of disconnected nodes (nodes with no edges), and connected components (nodes with greater than or equal to one edge).

## Serum collection

Animals were anesthetized using isoflurane and remained anesthetized throughout the duration of serum collection. The animal was placed in a supine position and its paws were secured to spread out the torso. A small incision was made with scissors and forceps in the skin above the xiphoid process of the sternum and expanded to reveal the bottom of the rib cage. The peritoneum in the same area was then cut to reveal the underside of the diaphragm. The diaphragm was cut and cleared away with scissors, thus exposing the bottom of the heart. A clamp was used to hold the ribcage out of the way, and a 25 ½ G needle attached to a 1 mL insulin syringe was used to collect blood directly from the apex of the heart. Using this method, roughly 800 µL of blood could be collected from each animal. After collection, the animals were euthanized via cervical dislocation. The collected blood was allowed to clot in a 2 mL tube at room temperature for one-hour. Then, the blood was spun at 16,100 g for 15 min at 4°C in a microcentrifuge. The serum was collected and aliquoted into 50 µL tubes and stored at −20°C until used.

## Circulating Klotho and FGF23 quantification

Blood collected from experimental and control animals was clotted at room temperature for 1 hr, after which it was spun at 16,100 G for 15 min in a fixed rotor centrifuge at 4°C in order to separate

serum. This serum was diluted 1:25 and 1:6 in diluent buffer to quantify Klotho (Cloud Clone Corp), and FGF23 (Abcam), respectively using commercially available ELISA kits.

## Klotho ELISA

The Klotho ELISA was conducted according to the protocol using the Cloud Clone Corp. ELISA Kit for mouse Klotho (SEH757Mu, Lot: L180223640). This kit was validated by *Sahu et al., 2018*. Briefly, blood serum samples and standards were diluted in PBS (1:25) and 100 µL of the samples were added in duplicates to a 96-well plate pre-coated with a biotin-conjugated antibody specific for Klotho detection. This plate was then incubated at 37°C for 1 hr. Samples and standards were removed and 100 µL of biotin-conjugated antibody cocktail was added to each well and incubated at 37°C for 1 hr. The microplates were washed three times with washing buffer provided by the kit (2 min per wash), using an automated plate washer (BioTek 50TS). Following this, 100 µL of Avidin conjugated Horseradish-Peroxidase was added to each well and incubated at 37°C for 30 min. The plate was washed with buffer five times, following which 90 µL of tetramethylbenzidine (TMB) substrate was added to each well and the plate was incubated at 37°C for 20 min. A sulfuric acid-based stop solution was added to terminate the reaction and optical density of each well was measured with a Spectramax M3 plate reader (Molecular Devices) at a wavelength of 450 nm. The analysis was conducted using Microsoft Excel by generating a 2P equation for concentration based on the standard curve and plotting the averaged absorbance values for each sample along this curve.

Any sample displaying evidence of hemolysis was not included in data analysis, as we have seen that hemolysis interferes with values. Samples were not subjected to multiple free-thaw cycles, as we have observed that this may affect α-Klotho detection (*Cheikhi et al., 2019*).

## FGF23 ELISA

The FGF23 ELISA was performed according to the protocol provided by the manufacturer (Abcam, ab213863, Lot: GR3326863). Briefly, blood serum samples and standards (diluted 1:6 in sample diluent buffer) were added in duplicates to a 96-well plate pre-coated with antibody against FGF23 and incubated for 90 min at 37°C. The plate content was then discarded and the plate was blotted. Following this, a biotinylated anti-mouse FGF23 antibody was added to all wells and incubated for 60 min at 37°C. Next, the plate was washed three times with 0.01M PBS using the same plate washer as described above. A total of 100 µL of avitin-biotin-peroxidase complex was added to each well and incubated again for 30 min at 37°C. The plate was washed five more times, following which 90 µL of TMB substrate was added to each well and incubated in the dark for 20 min at 37°C. Finally, 100 µL of TMB stop solution was added and the plate was immediately read at 450 nm using the same plate reader as described above. Analysis was conducted as described above. Hemolyzed samples were not included, and each sample was frozen once prior to the experiment.

Eleven of the samples returned absorbance values lower than the lowest value of the standard curve. Rather than extrapolating and recording negative concentration values for these samples (which would not be biologically coherent), we assumed that these samples did not have detectable levels of FGF23. Thus, their concentration was recorded as zero. These samples have been marked with red symbols in the figure.

## Meso-Scale discovery (MSD) Klotho ELISA

Detection of circulating Klotho levels from mouse blood samples was performed by using an ELISA assay on the MSD platform. Standard-bind MSD plates (#L15XA-1, MSD) were coated with mouse Klotho capture antibody (#AF1819, R and D) at a final concentration of 4 µg/mL in PBS for 1 hr at room temperature. Subsequently, plates were washed three times using wash buffer (PBS + 0.05% Tween-20) at 300 µL/well followed by an incubation with 3% blocker A solution (R93BA-2, MSD) for 1 hr at room temperature. Dilutions of serum samples and murine recombinant Klotho standard (#1819 KL-050, R and D) were prepared in 1% blocker A solution and added to the MSD plate in a final volume of 25 µL/well after the plate was washed three times. Samples were incubated for 1 hr at room temperature followed by three washing steps. Detection antibody (#BAF1819, R and D) was diluted to 1 µg/mL in 1% Blocker A solution in PBS and SULFO-tag labeled streptavidin (#R32AD-5, MSD) was diluted to 0.5 µg/mL in 1% Blocker A in PBS. Both dilutions were added to the plate simultaneously (25 µL/well each) and incubated for 1 hr at room temperature. After three washing

steps, 1x Read Buffer (#R92TC-2, MSD) diluted in water was added at 150 µL/well. Electrochemiluminescence was detected in the MSD Sector Imager 600.

## Metabolite analysis

Serum metabolites were analyzed by the Mouse Metabolic Phenotyping Center at the University of Cincinnati.

## AAV production

Recombinant AAV8 was used as vector for expressing murine Klotho or enhanced green fluorescent protein (GFP). The genomic constructs contained the LP-1 promoter, GFP or a codon-usage optimized sequence of murine alpha klotho (Geneart, Regensburg, Germany) with an N-terminal secretion signal originating from human CD33 (sequence: MPLLLLLPLLWAGALA) and a C-terminal V5-tag epitope (*Southern et al., 1991*), followed by a WPRE sequence and an SV40 polyA signal. The expression cassettes were flanked by AAV2-derived inverted terminal repeats. The recombinant AAV8-LP1-mKlotho and AAV8-LP1-eGFP vectors were prepared as previously described (*Strobel et al., 2019*). Briefly, human embryonic kidney cells (HEK-293H cells, Thermo Fisher Scientific) were cultured in Dulbecco's modified Eagle's medium +GlutaMAX I+10% fetal calf serum (Gibco/Thermo Fisher Scientific) and transfected as previously described (*Strobel et al., 2015*). AAV purification via polyethylene glycol precipitation, iodixanol gradient, ultrafiltration, and sterile filtering was conducted as described previously (*Strobel et al., 2019*). Genomic titers of purified AAV8 vector stocks were determined by isolation of viral DNA and subsequent qPCR analysis using primers specific for the LP-1 promoter with the following primer sequences; forward: GACCCCC TAAAATGGGCAAA. reverse: TGCCCCAGCTCCAAGGT.

## AAV tail vein injection

Aliquots of AAV-Kl (1.79e13 Vg/ml) and AAV-GFP (3.48e12 Vg/ml) were stored in −80℃ and thawed within an hour of use. Animals were restrained using a tail illuminator restrainer from Braintree Scientific Inc and injected with either a GFP expressing non-targeting control or AAV-Kl via the tail vein using a 31G insulin syringe. Viral solutions were diluted in Dulbecco's phosphate-buffered saline (DPBS) to a final volume of 100 µL/mouse. A successful injection entailed full administration of the viral solution into the vein smoothly, without swelling or bleeding following syringe removal. Only successfully injected mice were included in the study (a total of six were excluded).

## AAV-Klotho in vivo dose response study

Female C57Bl/6 mice (9–12 weeks old) with a body weight of 19–21 g were purchased from Charles River Laboratories. AAVs were diluted to the desired concentrations in Dulbecco's phosphate-buffered saline (DPBS) and administered into the tail vein under light isoflurane anesthesia. The doses tested were $1 \times 10^8$, $3 \times 10^9$, and $1 \times 10^{10}$ vg/mouse AAV-Kl compared with $1 \times 10^{10}$ AAV-GFP control. The final volume for injection was 100 µL per mouse. Three weeks after injection with AAV-GFP, AAV-Kl or DPBS, animals were sacrificed and blood samples were collected for the detection of circulating Klotho levels by MSD-ELISA. Livers were also collected to quantify AAV vector genomes. For vector DNA isolation flash-frozen liver samples, were homogenized in 900 µL of RLT buffer (79216, Qiagen), using a Precellys-24 homogenizer and ceramic bead tubes (KT03961-1-009.2, VWR) at 6000 rpm for 30 s. Homogenates were further processed by applying standard Phenol-chloroform extraction. Finally, vector DNA was purified by using the AllPrep DNA/RNA 96 kit (80311, Qiagen) according to the manufacturer's instructions. Detection of AAV vector genomes was performed by qPCR using primers specific for the LP-1 promoter (forward: GACCCCCTAAAA TGGGCAAA. reverse: TGCCCCAGCTCCAAGGT). For quantification, a standard curve generated by serial dilutions of the respective Klotho expression plasmid was used. qPCR runs were performed on an Applied Biosystems ViiA 7 Real-Time PCR System. Animal experiments in this study were approved by the local German authorities (Regierungs-präsidium Tübingen) and conducted in compliance with the German and European Animal Welfare Acts.

## AAV-administration, injury, and sarcopenia experimental models

For the muscle regeneration experiment, the dose of GFP and AAV-Kl used was $1 \times 10^{10}$ vector genomes(vg) per animal. Five days following injection, each mouse received a bilateral intramuscular injury to the tibialis anterior muscle (TA) using 10 µL of 1 mg/mL cardiotoxin (*Naja pallida,* Sigma, molecular weight 6827.4), and animals were euthanized 14 days after injury. In uninjured animals, the dose used was $3 \times 10^{8}$ vector genomes per mouse, and mice were euthanized 14 days after AAV injection.

## Flow cytometry

To quantify the ratio of fibroadipogenic progenitor cells (FAPs) to MuSCs (MuSCs) following muscle injury, old mice were injected with GFP or AAV-Kl at $1 \times 10^{10}$ vg/animal (9 animals per group) via tail vein. As with the previous injury experiment, animals received a bilateral cardiotoxin injury to the TA 5 days following AAV injection. After 3 days, animals were euthanized via cardiac puncture, and the TA muscles from each animal were collected and digested according to a previously established protocol for flow cytometry (*Yi and Rossi, 2011*; *Liu et al., 2015*). Though we began with nine mice in each group, we pooled together sets of TAs from 3 mice to ensure an adequate amount of tissue for digestion and sorting . Thus, the final sample size for each group was n = 3 (3 pools of 3 sets of TAs). Following digestion, FAPs (CD31⁻, CD45⁻, Sca1⁺, $\alpha-7^-$) were sorted from MuSCs (CD31⁻, CD45⁻, Sca1⁻, $\alpha-7^+$) via fluorescence-activated cell sorting (FACS). The ratio of FAPs:MuSCs was quantified and recorded for each experimental replicate.

## Exercise-induced injury

To determine how muscle administration of AAV-Kl affects adaptability to exercise, we adopted the eccentric exercise protocol created by *Armand et al., 2003*. Mice were acclimated to treadmill (Columbus Instruments) running over 4 days. On day 1 they ran at 6 m/min for 20 min. Day 2 was at 8 m/min for 20 min. Day 3 was at 8 m/min for 20 min on a 15-degree decline. Day 4 was at 10 m/min for 20 min on a 15-degree decline. The mice then were given 1 day of rest after acclimation before the eccentric injury protocol, which consisted of running at 10 m/min for 60 min on a 15-degree decline. A slight electric shock and gentle prodding were used to ensure adherence. We also assessed exercise protocol adherence according to a previously established method (*Ríos-Kristjánsson et al., 2019*) to ensure equal performance between groups. Mice that scored below 0.3 were excluded from further analysis due to their inability to complete the injury protocol. Other studies used longer runs and faster speeds to induce injury, but these previous studies were conducted in young mice. Old mice were not capable of running at such speeds and times. Still, to confirm the efficacy of the protocol, histology was conducted to visualize the extent of injury.

On the first day of acclimation, mice were randomized into two groups (n = 6 per group, repeated in two cohorts), one receiving AAV-Kl at $1 \times 10^{10}$ vg per mouse and one receiving the GFP control via tail vein injection as described above. The timing of administration was decided so that AAV injection would fall 5 days prior to muscle injury, similar to our cardiotoxin injury model. Seven days after the muscle injury protocol, contractile testing was conducted, mice were euthanized, and serum and TA muscle were collected. One mouse in the AAV-Kl group was removed from analysis for maximum specific tetanic force production due to an error in testing. TA muscles were sectioned, stained for DAPI and laminin, and imaged at 10x using a Zeiss Observer Z1. Centrally nucleated fibers were quantified using the MuscleJ macro (*Mayeuf-Louchart et al., 2018*). The mean value for each sample was calculated from three separate images, and the mean percentages were compared between the mice that had undergone the eccentric protocol vs. uninjured control mice.

## Statistical analysis

Analyses were performed using GraphPad Prism version nine software. Shapiro-Wilk and Levene's tests were initially performed to assess normality of data and equality of variances, respectively. If assumptions of normality and homogeneity of variances were met, a Student's t-test was performed while comparing two groups, or a one-way analysis of variance (ANOVA) test with Tukey post-hoc comparison. When conditions for normality were not met in two group comparisons, the groups were compared using Mann-Whitney U test. A Welch's test was applied when there were differences between the standard deviations of two groups. When conditions of normality or homogeneity were

not met for comparisons of three or more groups, a Kruskal-Wallis test with Dunn's post-hoc comparison was used All results were expressed as mean ± standard deviation. Statistical significance was established, a priori, at $p \leq 0.05$.

## Acknowledgements

The authors thank Matthias Duechs (Boehringer-Ingelheim (BI)) who performed MSD-ELISA on samples obtained from the dose testing experiment in young animals. We thank Timothy Lezon, Assistant Professor, Computational and Systems Biology at the University of Pittsburgh for valuable discussions regarding Shannon entropy and network analysis. We also thank Center for Research and Computing at the University of Pittsburgh that provided resources for network entropy computations. Amanda Miller, University of Pittsburgh, contributed to histology and flow cytometry. We are grateful to the Bryan Brown Lab, University of Pittsburgh, for equipment use and technical guidance. Finally, we thank Center for Biologic Imaging, University of Pittsburgh for providing resources to perform histological IF staining. The studies reported in this manuscript are supported by funding from NIA R01AG052978 (FA), NIA R01AG061005 (FA), and BI (FA).

## Additional information

### Competing interests

Joerg D Hoeck, Sebastian Kreuz: is an employee of Boehringer INgelheim Pharmaceutical Company. Michael Franti: is an employee of Boehringer Ingelheim Pharmaceutical Company. The other authors declare that no competing interests exist.

### Funding

| Funder | Grant reference number | Author |
| --- | --- | --- |
| National Institute on Aging | R01AG052978 | Fabrisia Ambrosio |
| National Institute on Aging | R01AG061005 | Fabrisia Ambrosio |
| Boehringer Ingelheim | | Fabrisia Ambrosio |

The funders had no role in study design, data collection and interpretation, or the decision to submit the work for publication.

### Author contributions

Zachary Clemens, Sruthi Sivakumar, Abish Pius, Data curation, Formal analysis, Visualization, Methodology, Writing - original draft, Writing - review and editing; Amrita Sahu, Data curation, Formal analysis, Visualization, Methodology, Writing - review and editing; Sunita Shinde, Nathaniel Luketich, Data curation, Formal analysis; Hikaru Mamiya, Data curation; Jian Cui, Formal analysis; Purushottam Dixit, Methodology, Writing - review and editing; Joerg D Hoeck, Sebastian Kreuz, Validation, Methodology; Michael Franti, Validation, Methodology, Writing - review and editing; Aaron Barchowsky, Conceptualization, Resources, Supervision, Funding acquisition, Writing - original draft, Project administration, Writing - review and editing; Fabrisia Ambrosio, Conceptualization, Resources, Supervision, Funding acquisition, Methodology, Writing - original draft, Project administration, Writing - review and editing

### Author ORCIDs

Zachary Clemens https://orcid.org/0000-0002-2230-9151
Sruthi Sivakumar https://orcid.org/0000-0003-0649-8551
Aaron Barchowsky http://orcid.org/0000-0003-1268-8159
Fabrisia Ambrosio https://orcid.org/0000-0002-5497-5968

## Ethics

Animal experimentation: All animal experiments were performed with prior approval from the Institutional Animal Care and Use Committee of the University of Pittsburgh. These experiments were conducted in accordance with protocol 17080802 (University of Pittsburgh ARO: IS00017744). All surgeries and invasive procedures were performed under isoflurane anesthesia, with painkillers administered afterwards. Every effort was made to minimize suffering.

## Decision letter and Author response

Decision letter https://doi.org/10.7554/eLife.61138.sa1
Author response https://doi.org/10.7554/eLife.61138.sa2

# Additional files

## Supplementary files

• Supplementary file 1. Full list of KEGG pathways affected by Klotho treatment in old and oldest-old animals.
• Supplementary file 2. Search terms and network properties for hallmarks of aging genes.
• Transparent reporting form

## Data availability

Sequencing data has been deposited in GEO accession: GSE156343.

The following dataset was generated:

| Author(s) | Year | Dataset title | Dataset URL | Database and Identifier |
|---|---|---|---|---|
| Clemens Z, Sivakumar S, Pius A, Sahu A, Cui J, Barchowsky A, Ambrosio F, Clemens Z, Sivakumar S, Pius A, Sahu A, Cui J, Barchowsky A, Ambrosio F | 2021 | The biphasic and age-dependent impact of Klotho on hallmarks of aging and skeletal muscle function | https://www.ncbi.nlm.nih.gov/geo/query/acc.cgi?acc=GSE156343 | NCBI Gene Expression Omnibus, GSE156343 |

The following previously published datasets were used:

| Author(s) | Year | Dataset title | Dataset URL | Database and Identifier |
|---|---|---|---|---|
| Schaum N, Lehallier B, Hahn O, Pálovics R, Hosseinzadeh S, Lee SE, Sit R, Lee DP, Losada PL, Zardeneta ME, Fehlmann T, Webber JT, McGeever A, Calcuttawala K, Zhang H, Berdnik D, Mathur V, Tan W, Zee A, Tan M, The Tabula Muris Consortium, Pisco AC, Karkanias J, Neff NF, Keller A, Darmanis S, Quake SR, Wyss-Coray T | 2020 | Tabula Muris Senis | https://tabula-muris-senis.ds.czbiohub.org/ | NCBI Gene Expression Omnibus, GSE132040 |
| Tumasian RA, Harish A, Kundu G, Yang JH, Ubaida- | 2021 | Skeletal Muscle Transcriptome in Healthy Aging | https://www.ncbi.nlm.nih.gov/geo/query/acc.cgi?acc=GSE164471 | NCBI Gene Expression Omnibus, GSE164471 |

| | | | | | |
|---|---|---|---|---|---|
| Mohien C, Gonzalez-Freire M, Kaileh M, Zukley LM, Chia CW, Lyashkov A, Wood WH, Piao Y, Coletta C, Ding J, Gorospe M, Sen R, De S, Ferrucci L. | | | | | |
| Schaum N, Lehallier B, Hahn O, Pálovics R, Hosseinzadeh S, Lee SE, Sit R, Lee DP, Losada PL, Zardeneta ME, Fehlmann T, Webber JT, McGeever A, Calcuttawala K, Zhang H, Berdnik D, Mathur V, Tan W, Zee A, Tan M, The Tabula Muris Consortium, Pisco AC, Karkanias J, Neff NF, Keller A, Darmanis S, Quake SR, Wyss-Coray T | 2019 | Age-Related Gene Expression Signature in Rats Demonstrate Early, Late, and Linear Transcriptional Changes from Multiple Tissues | https://www.ncbi.nlm.nih.gov/bioproject/?term=PRJNA516151 | NCBI BioProject, PRJNA516151 |

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
