## [Decision Letter]

**Acceptance summary:**

The manuscript provides a comprehensive characterization of the trajectory of murine muscle aging and correlates that to the ability of Klotho to exert beneficial effects on muscle health. The novel entropy-based modeling defines the trajectory of murine muscle aging, which will certainly be of interest to the community of researchers studying the biology of aging.

**Decision letter after peer review:**

Thank you for submitting your article "The biphasic and age-dependent impact of Klotho on hallmarks of aging and skeletal muscle function" for consideration by *eLife*. Your article has been reviewed by 4 peer reviewers, and the evaluation has been overseen by a Reviewing Editor and Jessica Tyler as the Senior Editor. The following individual involved in review of your submission has agreed to reveal their identity: Daniel Remondini (Reviewer #3).

The reviewers have discussed the reviews with one another and the Reviewing Editor has drafted this decision to help you prepare a revised submission.

Summary:

The manuscript by Pius et al. seeks to comprehensively characterize the trajectory of murine muscle aging and to correlate that to the ability of Klotho to exert beneficial effects on muscle health. The authors present a comprehensive profiling of skeletal muscle aging at the histological, functional, and transcriptomic levels across the lifespan (young, middle, old, oldest old age) and network entropy as a measure of molecular dysfunction during aging. The authors identify a network entropy inflection point and find it correlate with the ability of Klotho delivery to attenuate age-related changes. Interestingly, network entropy does not further increase from old to oldest old and Klotho in oldest old fails to exert a benefit at that age. The study presents a large amount of work, particularly in compiling the various datasets and developing a novel entropy-based modeling to define the trajectory of murine muscle aging, which will certainly be of interest to the community of researchers studying the biology of aging. All four reviewers agreed that the study provides interesting insights into aging. However, as detailed below, various concerns were also raised, relating in large part to insufficient clarity in the current version with respect to the authors' methods and the limitations of their approach and data.

Essential revisions:

1. As the transcriptomes and network entropy analysis are the cornerstone of the authors' approach, it is essential that they provide clear and complete details regarding the experimental strategies and rationale underlying each analysis. In particular, detailed answers to the following key questions must be available in the present manuscript:

2. There were mixed reviews on the validity and advantage of the entropy network method: two positive and two negative reviews. The entropy network model was built on protein-protein interaction (PPI) networks. Given that the authors used transcriptomes as input data, provide the validity of the methodology and discuss the limitation of the approach. For example, numerous reports show that the lack of positive linear correlation between protein levels and the amount of mRNA, e.g. Cell 165: 535-550; Molecular and Cellular Proteomics 1: 304-313, Scientific Reports Article number: 3272.

3. The authors should provide justification as to why the entropy change from 522 to 528 (only 1.1%, p=0.07) in Figure 3D is deemed meaningful and important.

4. pg 7 line 123-130 and Figure 3. From the figure it emerges that the first component of PCA seems to characterize aging progression: which are the genes mostly associated to this component? Further analysis (e.g. gene selection based on loadings, possibly after Factor Analysis) could help to emphasize these aspects and identify the key genes associated to aging progression. Moreover, if biologically relevant, these gene set could be characterized throughout the study in relation to aging progression and to the effects of Klotho. Further: did authors try other dimensionality reduction methods such as t-SNE, that in principle could identify other peculiar aspects within the data?

5. Methods section: more details about the analyzed network are required, e.g. which is the size of the networks used to estimate network entropy? Which are the main global features (e.g. link density, presence of disconnected components,.…). Moreover, there appears to be more than one network that has been analyzed (genes associated to hallmarks of aging in line 146, global PPI network on line 155). A clear description is needed, in the Methods or in Supplementary material. Further: in Figure1D and supplementary Figure S1 the Network entropy values appear with a "confidence interval" (in light blue). In the paper there are no references about how this uncertainty was estimated: is it standard deviation or standard deviation of mean? It is a 95% confidence interval or something else? Also for this point a clear description is requested.

6. In Figure 3, the gene perturbation data presentation is a bit odd; what would be much more helpful would be heat maps showing the progression of gene changes from young to old to oldest-old, and which of these are counter-regulated by Klotho over-expression, in particular, for the pathways shown to be linearly regulated with age, but counter-regulated by Klotho in the old. Figure 3G is redundant. Remove 3G and instead indicate the DEG numbers in each pie chart in Figure 3F. Figure 6C is also redundant. As in the Figure 3D, 3E and 3F, in Supplemental Figure 2B Old vs Young should be compared to Oldest-old vs Young age group, instead of comparing Oldest-old vs Old. For oldest-old it would be interesting to see what Klotho does. The authors get at this in figure 6D but it would important to show the actual degree of perturbation connected to the pathways shown.

7. It seems like the oldest-old Klotho vs oldest-old pathways go in the opposite direct as Old Klotho vs Old. Is that true of every pathway? it would be helpful to connect the gene expression changes to a mechanism.

8. There is a substantial literature on the skeletal muscle transcriptional studies throughout the lifespan. An example in mice (https://doi.org/10.1016/j.biocel.2014.04.025), a very recent example in rats (https://doi.org/10.1016/j.celrep.2019.08.043) and in humans (https://journals.plos.org/plosgenetics/article?id=10.1371/journal.pgen.0020115). Refer to the previous work, which are missing in the current manuscript.

9. The idea that "molecular disorder" underpins age-related muscle dysfunction is clever. Assuming the validity of the approach, an obvious question is whether the "entropy" approach used in this manuscript can be applied to the available datasets to validate the work and to compare differences across species. This translational component and general applicability is important to show and will increase the impact of this work.

10. There is no reference about repository of the RNA-seq data, which is mandatory for *eLife*. Otherwise authors should justify their decision about why not to make data accessible.

11. Would it be possible to generate an interactive online website tool to probe this dataset (and perhaps the others above), which includes the "entropy" analysis of molecular disorderliness performed here? This could be very useful to the scientific community and reduces the concern that the authors are under-utilizing their novel approach to gene expression analysis.

12. The baseline characterizations of aging are well-done and consistent with prior studies. Likewise, the experiments involving the over-expression of Klotho do not appear novel. Detailed answers to the following key questions must be available in the present manuscript to provide a novelty of the study:

13. The progressive underexpression of Klotho (as shown in Figure 4A) is much smoother than the growth in FGF23 (Figure 4B) that in the paper is claimed to be the major interactor of Klotho: is there any explanation for that? Maybe the group of direct (or second neighbor) interactors of Klotho (and of FGF23 if it is relevant) should be studied specifically, to characterize the surrounding environment of these genes and provide a deeper view of the possible processes associated to aging and to Klotho response. Moreover, it is shown a large difference between growth of FGF23 in oldest-old female as compared to male of the same age group (in female mice the growth starts earlier): can the authors try to explain why? (Maybe FGF23 interactome could provide further insight?)

14. More information on the effectiveness of the AAV is needed. What organs are being affected by this? How much over-expression of Klotho is there with AAV-Kl, and specifically how much in muscle? Explain why the AAV approach was used instead of Klotho protein supplementation, which seems to have been effective in the past and is a safer and more translatable approach.

15. The effects of Klotho are interesting; what would have been helpful would be some pharmacodynamic/pathway studies. Are the effects seen happening as a result of increased FGFR signaling? FGF23 levels go up with age, while Klotho levels go down. This may be a reaction to low FGFR signaling. When Klotho is given back, which are the key pathways that are noted to change? In other words, it would be helpful to connect the gene expression changes to a mechanism.

16. As a comment in the final discussion, since the results about mice treated with Klotho are stated as general, but transcriptomics profiling was performed only on female samples, it should be emphasized that these results could not be generalized independently of sex, since many differences have been observed between the two genders.

17. Following relevant issues emerge that require further analysis, that should not alter the overall results of the paper but in my opinion could reinforce them. Detailed answers to the following key issues must be provided, including justification as to why suggested experiments cannot be done:

18. Page 10 line 215; AAV-KL treatment rather decreased force in oldest-old group (Figure 5F and 5G). Given that Klotho over-expression in the oldest-old is perhaps detrimental, experiments should be undertaken to explore how increasing Klotho earlier in life influences regenerative and transcriptional responses later in life. If the positive attributes of Klotho, when applied to an "unhealthy" oldest-old environment, then become a negative stress, it is an idea that should be explored further. It would also be welcome to see how Klotho influences more translatable outcomes such as adaptability to exercise training, since the link between regenerative potential and age-associated sarcopenia is unclear.

19. For the mouse muscle IHC phenotyping, it is warranted to show other measures such as fiber type-specific CSA, quantification of ECM (via Sirius Red or Masson's), markers of denervation, as well as intramuscular adipose tissue infiltration (IMAT). Extending the analysis to a more oxidative/slow-twitch muscle such as the soleus would also be welcome, given that the TA mostly contains muscle fibers that aren't present in humans (Type IIB).

20. Further characterization of the AAV-Klotho regeneration experiments is warranted. Why is there less fibrosis in AAV-Kl? Are there fewer FAP/fibrogenic cells? Are there more satellite cells? Presenting collagen 4 intensity (Figure 4H) seems unorthodox. Please show Sirius Red or Masson's and normalized to area. In the aged AAV-Kl experiments, why is there less lipid accumulation, which appears to be specifically within (and not between) the muscle fibers? Was lipid accumulation between muscle fibers (IMAT) different? An Oil Red O analysis is warranted.

21. Serum levels of cholesterol and insulin of young mice (3-6 month) is necessary in Figure 4D to compare with those of old mice overexpressing Klotho.

22. As previously reported by the authors in Nature communication (2018), it would be interesting to compare the Klotho's function in the mitochondria of muscle progenitor cell, with the Klotho's function here identified.

23. To verify the author's suggestion in Discussion section, it is important to examine whether F2 or Kng2 plays a bifurcation role in old and oldest muscles.

24. The key experiment that seems to be missing is the manipulation of Klotho earlier in life to try and prevent the onset of "molecular disorder" later in life, especially since acute Klotho over-expression in the oldest old seemed to have the opposite effect.

25. A summary figure that highlight the main findings of this manuscript would be a welcome addition.

26. Abstract has no conclusions and does not properly reflect contents of the results.

---

## [Author Response]

Essential revisions:1. As the transcriptomes and network entropy analysis are the cornerstone of the authors' approach, it is essential that they provide clear and complete details regarding the experimental strategies and rationale underlying each analysis. In particular, detailed answers to the following key questions must be available in the present manuscript:2. There were mixed reviews on the validity and advantage of the entropy network method: two positive and two negative reviews. The entropy network model was built on protein-protein interaction (PPI) networks. Given that the authors used transcriptomes as input data, provide the validity of the methodology and discuss the limitation of the approach. For example, numerous reports show that the lack of positive linear correlation between protein levels and the amount of mRNA, e.g. Cell 165: 535-550; Molecular and Cellular Proteomics 1: 304-313, Scientific Reports Article number: 3272.

This is a valuable discussion point, and we are grateful for the opportunity to clarify the use of the PPI in our paper. We agree that the nonlinear relationship between gene expression and protein levels precludes one-to-one inferences regarding proteomic muscle changes over time. However, our goal was not to estimate protein levels from transcriptomic data. Instead, we use the PPI network as a means to quantify changes in the interactome of age-related genes. To do this, we map skeletal muscle RNA-seq data using experimentally-validated physical interactions. The advantage of this approach is that it provides a proxy to visualize the functional relationship between genes. On the other hand, edges in the gene co-expression network are based on solely correlation cutoffs, and, hence, may not necessarily represent meaningful interactions. As a result, we did not view the use of a gene co-expression network as an appropriate approach. Previous reports have similarly used the PPI interactome to elucidate how shifts in gene expression network properties may play a role in the onset of complex diseases (Banerji, Miranda-Saavedra et al. 2013, Wu, Xie et al. 2014, Yang, Zhang et al. 2016, Teschendorff and Enver 2017, Fan, Lin et al. 2018).

Despite the advantage of a PPI approach, we acknowledge limitations. Namely, in addition to the fact that mapping of genes to proteins is not one-to-one, there are also a number of proteins for which PPI information has not been documented. We have highlighted both the rationale of our approach as well as its limitations in the Discussion (lines 343-349).

3. The authors should provide justification as to why the entropy change from 522 to 528 (only 1.1%, p=0.07) in Figure 3D is deemed meaningful and important.

This is a particularly intriguing question considering others have similarly observed that entropy changes in the context of stem cell differentiation, cancer, and aging are quite subtle (Park, Lim et al. 2016, Teschendorff and Enver 2017, Kannan, Farid et al. 2020).

From a mathematical perspective, it is not surprising that the change in entropy over time is relatively small given that the spatial network entropy represents the total number of possible configurations (i.e., flexibility of the system), which has a magnitude on the order of 10^6^. The entropy of the network is a reflection of the network architecture, which is dependent on the number of nodes, the number of edges, and the edge weights (i.e., the difference in gene expression). Since fundamental aspects of the network architecture, including the number of nodes and the number of edges, remains relatively stable over time, the change in the magnitude of network entropy represents only a small fraction of all possible configurations (~1.1%). However, this small percentage captures the change in expression of a large number of genes (~7600 genes), each of which have the potential to elicit a cascade of downstream phenotypic changes.

From a physiological perspective, there is growing evidence that very small-scale mRNA changes can lead to extreme changes in protein expression and cellular phenotype (Ruzycki, Tran et al. 2015, Cheng, Teo et al. 2016). In the context of cancer, for example, a single base mutation can drive extensive cellular remodeling that extends well beyond the predicted downstream responses, a so-called ‘butterfly effect’ (Hart, Zhang et al. 2015). Others have similarly shown that cells are highly sensitive to initial conditions (in our case, gene expression), and that very small changes in transcriptional regulation and equilibrium may exert potent downstream impacts on disease progression (Dorn 2013, Desi and Tay 2019). We have revised the manuscript to highlight these important points (lines 353-368).

4. pg 7 line 123-130 and Figure 3. From the figure it emerges that the first component of PCA seems to characterize aging progression: which are the genes mostly associated to this component? Further analysis (e.g. gene selection based on loadings, possibly after Factor Analysis) could help to emphasize these aspects and identify the key genes associated to aging progression. Moreover, if biologically relevant, these gene set could be characterized throughout the study in relation to aging progression and to the effects of Klotho.

Great suggestion. To address this, first, we evaluated the loading of genes for PC1 and PC2. Interestingly, functional enrichment of top 25 genes associated with each component revealed that 9/25 genes (36%) in PC1, and 17/25 genes (68%) in PC2 are associated with hallmarks of aging. This is consistent with the fact that PC2 best distinguishes age groups (Figure 3B).

We then focused on the top 100 PC2 genes and performed functional gene ontology enrichment analysis (presented in Figure 3—figure supplement 1A). These GO terms were predominantly associated with altered intercellular communication and nutrient sensing deregulation, two of the hallmarks of aging.

Next, we visualized the top 100 genes from PC2 loadings as a heatmap. The heatmap showing all gene trends is presented in Figure 3—figure supplement 1B. Fifteen genes have an increasing trend with aging progression, and 23 genes had a decreasing trend. A heatmap focusing only on these 38 up- and down-regulated genes is presented in Figure 3C. Nutrient sensing deregulation and altered intercellular communication were the two hallmarks most highly associated with the 38 genes displaying an aging progression trend. We have updated the Results sections to present these findings (lines 138-143).

Interestingly, the two hallmarks that emerge different with PC2 genes: altered intercellular communication and nutrient-sensing deregulation also relates to FGF23 signaling that is a known interactor of Klotho. We observed a novel age-dependent effect on FGF23 signaling with Klotho treatment. We have discussed this effect in comments 13 and 15 in greater detail.

Further: did authors try other dimensionality reduction methods such as t-SNE, that in principle could identify other peculiar aspects within the data?

As suggested, we performed t-SNE to complement the PCA presented in Figure 3B. We were encouraged to see a similar clustering pattern for the aging samples (Author response image 1). Specifically, as was seen with the PCA, old and oldest-old displayed some degree of overlap with each other but were distinct from young muscles. Because the information gained by the two-dimensional reduction approaches were similar, we did not include the t-SNE map in the revised paper.

5. Methods section: more details about the analyzed network are required, e.g. which is the size of the networks used to estimate network entropy? Which are the main global features (e.g. link density, presence of disconnected components,.…). Moreover, there appears to be more than one network that has been analyzed (genes associated to hallmarks of aging in line 146, global PPI network on line 155). A clear description is needed, in the Methods or in Supplementary material. Further: in Figure1D and supplementary Figure S1 the Network entropy values appear with a "confidence interval" (in light blue). In the paper there are no references about how this uncertainty was estimated: is it standard deviation or standard deviation of mean? It is a 95% confidence interval or something else? Also for this point a clear description is requested.

We have now added Supplementary file 2 that describes the number of nodes, number of edges, number of connected components, link density, and spatial network entropy before normalization. The table 2 within the file is divided into several parts: (A) describes the network features for hallmark of aging associated genes presented in the main figure (Figure 3E), and (B) describes network features for Figure 3—figure supplement 2, which is calculated from all genes. Clarification about this in the main text is added (lines 666-670).

The figure legends have been revised to denote that light blue shaded portion represents the standard deviation.

Supplementary file 2 also has features for networks that were generated for male mice, rats and humans (2C, 2D and 2E respectively), later described in response to comment 9.

6. In Figure 3, the gene perturbation data presentation is a bit odd; what would be much more helpful would be heat maps showing the progression of gene changes from young to old to oldest-old, and which of these are counter-regulated by Klotho over-expression, in particular, for the pathways shown to be linearly regulated with age, but counter-regulated by Klotho in the old. Figure 3G is redundant. Remove 3G and instead indicate the DEG numbers in each pie chart in Figure 3F. Figure 6C is also redundant. As in the Figure 3D, 3E and 3F, in Supplemental Figure 2B Old vs Young should be compared to Oldest-old vs Young age group, instead of comparing Oldest-old vs Old.

We characterized aging progression genes by extracting genes with highest PC2 loadings and displayed this as a heatmap (described above in response to comment 4 and Figure 3—figure supplement 1). We did not see any striking trend in the aging progression genes with Klotho over-expression. However, as stated above, we observed an age-dependent effect on FGF23 signaling with Klotho treatment. We have discussed this effect in greater detail comments 13 and 15. We have suggested future studies to explore the relationship (Discussion, lines 400-412).

As per the reviewer’s suggestion, we have expressed the DE gene count on the network plot (current Figures 3G, 7B). We have removed the redundant figures (previously Figures 3F, and 6C), and the revised figures have been added as Figures 3G, and 7B.

Figure 3—figure supplement 3B (previously Supplemental figure 2B) has now been changed to show Old vs Young and Oldest-old vs Young.

For oldest-old it would be interesting to see what Klotho does. The authors get at this in figure 6D but it would important to show the actual degree of perturbation connected to the pathways shown.

We apologize for not being clear in the text and figure legend. Figure 7D (previously, Figure 6D) shows gene perturbation as total accumulation in the bar graph. The top pathways are perturbed in opposite directions in old and oldest-old. We have clarified this in the figure legend (line 232).

7. It seems like the oldest-old Klotho vs oldest-old pathways go in the opposite direct as Old Klotho vs Old. Is that true of every pathway? it would be helpful to connect the gene expression changes to a mechanism.

In the previous Figure 6C (current Figure 7C), we originally highlighted only those genes that displayed opposing responses to Klotho intervention when comparing old and oldest-old groups. However, not every pathway goes in the opposite direction. We have now expanded Supplementary file 1 (previously, supplemental figure 5) to present every pathway.

In order to probe potential mechanisms emerging from alterations in gene expression across the groups, we first performed gene ontology enrichment for the differentially expressed genes obtained after Klotho treatment (Figure 7A). Interestingly, we found that FGF/ion transport/calcineurin/NFAT signaling were amongst the top GO terms for old vs old klotho, but not oldest-old vs oldest-old klotho (Figure 7C, Table 1). We describe this age-dependent mechanism in greater detail in response to comments 13 and 15.

We next evaluated genes associated with the top 3 pathways differentially responsive to Klotho according to age: regulation of actin cytoskeleton, cGMP-PKG signaling pathway, and sphingolipid signaling pathway. In old, but not oldest old, mice, the majority of genes comprising these pathways were associated with calcium ion transport and signaling. These are well-established pathways that are regulated by Klotho (Kuro-o 2019). However, the age-dependency of these pathways in response to Klotho has not been shown. Instead, we found that the major responses to Klotho in the oldest-old group related to mitochondrial and metabolism associated processes (Figure 7C, Table 2). We present the top 20 functional gene ontology terms in Figure 7—figure supplement 2. We have also revised the Discussion accordingly (lines 400-412).

8. There is a substantial literature on the skeletal muscle transcriptional studies throughout the lifespan. An example in mice (https://doi.org/10.1016/j.biocel.2014.04.025), a very recent example in rats (https://doi.org/10.1016/j.celrep.2019.08.043) and in humans (https://journals.plos.org/plosgenetics/article?id=10.1371/journal.pgen.0020115). Refer to the previous work, which are missing in the current manuscript.

These studies are now cited in the Discussion (lines 337-340).

9. The idea that "molecular disorder" underpins age-related muscle dysfunction is clever. Assuming the validity of the approach, an obvious question is whether the "entropy" approach used in this manuscript can be applied to the available datasets to validate the work and to compare differences across species. This translational component and general applicability is important to show and will increase the impact of this work.

This is a great idea. To compare network entropy changes over time, as suggested, we accessed archived transcriptomic datasets from male mice, as well as rats and humans.

The male mice transcriptomic data was obtained from Tabula Muris Senis limb muscle (Schaum, Lehallier et al. 2020). We excluded females due to insufficient sample size. Age grouping was done as in our manuscript, with 3-6 months as young, 12-15 months as middle-aged, 21-24 months as old, and 27 months as oldest-old. The spatial entropy obtained from limb muscle sequencing is presented in Figure 3—figure supplement 4A. Network properties are presented in Supplementary file 2 (Table 2C). The blue shaded region denotes standard deviation (n=6, 7, 6, and 4 respectively). Unlike females, we saw that entropy continued to increase from old to oldest-old timepoints, which we found interesting considering that male mice showed less severe sarcopenic declines at the oldest-old age. Also interesting, entropy was lowest at the middle-aged timepoint, then gradually increased.

The rat spatial entropy is obtained from the dataset provided in comment 8, gastrocnemius sequenced from aging male rats ranging from young to middle-aged to old. The blue shaded region denotes standard deviation (n=7, 7, 8, 7, 7, 8, and 8 respectively) (Figure 3—figure supplement 4B). (Shavlakadze, Morris et al. 2019). Network properties are presented in Supplementary file 2 (Table 2D). These youngest and oldest age groups are roughly equivalent to 20, and 55 years in humans, respectively (Tidball et al., Experimental Gerontology, 2021). RNA-seq data was not available at the oldest-old timepoint (the average lifespan of *Rattus Norvegicus* (Sprague Dawley rat) used in this study is close to three years). Since we had more time points in rats, we observed entropy decreased from young to middle-age, after which time it increased into old age, consistent with our mice data.

For humans, the dataset referenced in comment 8 was not available, and the corresponding author of the study did not respond to our request for the data . As an alternative, we used RNA-seq data from a recent unpublished study on aging skeletal muscle obtained from *vastus lateralis* (GSE164471) (Tumasian RA 2021). Similar to the mice and rat trajectory, in humans, entropy decreased from young adulthood into the fourth decade of life, after which time it increased, reaching a tipping point in the sixth decade of life, then decreasing slightly. This is consistent with studies that show that sarcopenic declines begin in the fourth decade of life (Walston 2012, Yazar and Olgun Yazar 2019). PPI network entropy obtained from all genes in human males with ages ranging from young to middle-aged to old to oldest-old, grouped as decades, namely, 20-29 years, 30-39 years, and so on till >80 years old. We did not present females because of insufficient sample size. The blue shaded region denotes standard deviation (n=4, 7, 4, 5, 4, 7, and 4 respectively) (Figure 3—figure supplement 4C). Network properties are presented in Supplementary file 2 (Table 2E). Entropy trend is generally consistent with our mouse data. We have described this is Results (lines 176-187).

10. There is no reference about repository of the RNA-seq data, which is mandatory for eLife. Otherwise authors should justify their decision about why not to make data accessible.

We apologize for the oversight. The GEO database ID, GSE156343, has been added to the methods sections (line 607).

11. Would it be possible to generate an interactive online website tool to probe this dataset (and perhaps the others above), which includes the "entropy" analysis of molecular disorderliness performed here? This could be very useful to the scientific community and reduces the concern that the authors are under-utilizing their novel approach to gene expression analysis.

As recommended, we have created a web-platform that allows for easy estimation of network entropy using both transcriptomic and proteomic datasets: https://network-entropy-calculator.herokuapp.com/. The steps for preprocessing have been described in (https://github.com/sruthi-hub/sarcopenia-network-entropy). We added this link to the main text (lines 185-187 and Key Resources Table).

12. The baseline characterizations of aging are well-done and consistent with prior studies. Likewise, the experiments involving the over-expression of Klotho do not appear novel. Detailed answers to the following key questions must be available in the present manuscript to provide a novelty of the study:

We agree that the characterization of murine muscle aging is not the most novel aspect of our paper. Still, we felt that it was important to first thoroughly characterize age- and sex-related changes in muscle phenotype and transcriptomic profile over time ranging from young to very old age (a characterization we could not find in the literature) so as to provide a solid foundation for the experimental studies that followed.

Along these lines, to the best of our knowledge, no studies to date have demonstrated the age-dependent ability of Klotho overexpression to attenuate sarcopenia, nor has previous work identified the biphasic gene expression profiles that accompany these disparate responses to Klotho intervention when comparing old and oldest-old mice. It is our goal that this information may provide insights into the potential therapeutic window of Klotho overexpression in the treatment of sarcopenia as well as potential molecular mechanisms that may underlie anabolic resistance to therapeutic intervention in older individuals.

13. The progressive underexpression of Klotho (as shown in Figure 4A) is much smoother than the growth in FGF23 (Figure 4B) that in the paper is claimed to be the major interactor of Klotho: is there any explanation for that? Moreover, it is shown a large difference between growth of FGF23 in oldest-old female as compared to male of the same age group (in female mice the growth starts earlier): can the authors try to explain why? (Maybe FGF23 interactome could provide further insight?)

To address this question, first, we increased the sample size used for the FGF23 analyses. Re-running the analyses with additional samples further confirmed the trend initially observed (Figure 4B).

As pointed out by the reviewer, the increase in FGF23 is not as smooth as the decline in Klotho over time, a disconnect that is driven primarily by the female data. Males, on the other hand, display a relatively steady increase with age that coincides with steadily decreasing Klotho levels. We note that approximately 40% of the female population express exceedingly high levels of FGF23 at oldest-old age, whereas the remaining mice display a more linear increases in FGF23 compared to the other age groups. We posit that this discrepancy may be a result of varied hormonal status in aged female mice. Indeed, it has been reported that 25-40% female mice undergo menopause in older age, but shortly after, spontaneously regenerate their follicles, making them hormonally more similar to pre-menopausal mice (Review, Brinton RD, *Endocrinology* 2012). Since estrogen levels directly regulate circulating FGF-23 in both humans and rodents (Ix et al., *Am J Kidney Dis* 2011; Saki et al., *Gynecologic Endocrinology and Reproductive Medicine* 2020), the variability in FGF-23 levels may be a result of some of the mice having a menopausal hormonal profile while others do not. Unfortunately, we do not have serum from these animals in order to measure estrogen levels. Instead, we have raised this important point in the text (Results, lines 202-207).

However, this does not fully clarify possible relationships/interactions between Klotho and FGF23. First, there is no linear interaction in transcriptional regulation of the two genes, as they are on separate chromosomes and are predominantly expressed in different tissues. However, FGF23 protein will increase in response to loss of Klotho or transgenic global overexpression of Klotho (Xiao et al., *JCI Insigh*t 2019). The important interaction between Klotho and FGF23 is at the protein level where they act as co-ligands for *FGFR1* (Chen, Liu et al. 2018). At the same time, Klotho reduces FGF23 interactions with FGFR4 (Han et al., *J Mol Cell Cardiol* 2020, Xiao et al., *JCI Insigh*t 2019). We discuss this interaction in the context of the observation that many of the genes in the GO terms that are decreased by AAV-Klotho in the old mice are related to inhibition of FGFR4/PLCγ/calcineurin/NFAT transcriptional program. This may mechanistically account for some, but not all, of the improvements of old muscle and reduced adiposity. On the other hand, top GO terms when comparing oldest-old and oldest-old Klotho were not related to inhibition of FGFR4/PLCγ/calcineurin/NFAT signaling. Instead, we saw no patterns in these GO terms, which is indicative of a dysregulated response Klotho intervention in the oldest-old animals. This has been added to Results (lines 285-294), and discussion (lines 400-412). These pathways also presented in Figure 7C.

Maybe the group of direct (or second neighbor) interactors of Klotho (and of FGF23 if it is relevant) should be studied specifically, to characterize the surrounding environment of these genes and provide a deeper view of the possible processes associated to aging and to Klotho response.

Primary interactors with Klotho include Fgf23 and Fgfr4, which did not display a change in mRNA levels following Klotho treatment. We also probed for effects of AAV-Klotho treatment on several interactors of FGF23 and Klotho (*Fgf6*, *Fgfr4*, *Fgf22*, *Dmp1*). Again, we observed no significant changes in the expression of these interactors (Figure 7—figure supplement 1C). This suggests that the mechanism of Klotho in modulating function in the context of sarcopenia may be independent of transcriptional regulation of FGF23.

However, when we looked at the GO terms for pathways which were differentially expressed with Klotho treatment in old vs. oldest-old age groups, the changes in old were consistent with an inhibition of non-canonical FGF23 signaling through FGFR4/calcineurin/NFAT (Han, Cai et al. 2020). Importantly, this inhibition was not observed in the oldest-old age group. These results are presented in Figure 7C.

14. More information on the effectiveness of the AAV is needed. What organs are being affected by this? How much over-expression of Klotho is there with AAV-Kl, and specifically how much in muscle? Explain why the AAV approach was used instead of Klotho protein supplementation, which seems to have been effective in the past and is a safer and more translatable approach.

The main target organs for AAV8-mediated gene delivery in mice are the liver, heart, and skeletal muscle. However, by using the liver-specific LP1 promoter, Klotho expression is limited to hepatocytes. Circulating Klotho levels of AAV-Kl treated animals were approximately 50-100-fold higher when compared to untreated or AAV-GFP treated control groups, as assessed by Klotho-specific ELISA measurements in serum samples (Figure 4E). To address the question of Klotho overexpression, we also quantified vector genomes in the liver to quantify expression and included this data in Figure 4D. Klotho expression was not expected in skeletal muscle due to the liver-specificity of the AAV construct. Nevertheless, we extracted Klotho expression in gastrocnemius from our RNA-seq data and confirmed that levels were unchanged with AAV-Kl treatment, seen in Figure 4F.

We selected an AAV approach in order to allow for robust and stable Klotho levels over a three-week time frame without the need for frequent re-administration of recombinant Klotho. Due to the observed challenges of recombinant Klotho production, which typically results in very low protein yields, thereby raising therapeutic obstacles. AAV8-based gene therapy circumvents this issue. Of note, similar approaches using various AAV capsid variants, including AAV8, are currently in advanced clinical trials in the context of hemophilia and glycogen storage diseases (George 2017, Kishnani, Sun et al. 2019), thereby underscoring the feasibility of AAV-based gene therapy approaches for the continuous expression of secreted therapeutic proteins. We have added a rationale for the use of AAV in the discussion (Page 17, lines 382-386).

15. The effects of Klotho are interesting; what would have been helpful would be some pharmacodynamic/pathway studies. Are the effects seen happening as a result of increased FGFR signaling? FGF23 levels go up with age, while Klotho levels go down. This may be a reaction to low FGFR signaling. When Klotho is given back, which are the key pathways that are noted to change? In other words, it would be helpful to connect the gene expression changes to a mechanism.

As a first step to address the potential effects of Klotho administration on FGFR activity, we conducted an ELISA for FGF23 in the serum of the 21-24-, and 27–29-month-old female mice that had been treated with AAV-Klotho or GFP, as in Figure 6. We found no significant difference in FGF23 levels with Klotho treatment versus the age matched GFP treatment group (Figure 7—figure supplement 1A).

Klotho increases canonical FGF23 signaling by increasing the affinity of FGF23 to *FGFR1* by 20-fold (Ho and Bergwitz, *J Mol Endocrinol* 2021). At the same time, increased Klotho levels reduce interactions of FGF23 with FGFR4, which initiates the pathogenic calcineurin/NFAT transcriptional program (Ho and Bergwitz, *J Mol Endocrinol* 2021). Thus, the data suggest that, in old animals, the changes observed are not a result of increased FGFR signaling per se, but rather, a *change in the receptor* activated (FGFR4 to *FGFR1*) and subsequent downstream signaling by FGF23. These points are addressed in the results and in the discussion (lines 299-307, lines 400-412).

16. As a comment in the final discussion, since the results about mice treated with Klotho are stated as general, but transcriptomics profiling was performed only on female samples, it should be emphasized that these results could not be generalized independently of sex, since many differences have been observed between the two genders.

We have revised the text to highlight this limitation (lines 314-315).

17. Following relevant issues emerge that require further analysis, that should not alter the overall results of the paper but in my opinion could reinforce them. Detailed answers to the following key issues must be provided, including justification as to why suggested experiments cannot be done:18. Page 10 line 215; AAV-KL treatment rather decreased force in oldest-old group (Figure 5F and 5G). Given that Klotho over-expression in the oldest-old is perhaps detrimental, experiments should be undertaken to explore how increasing Klotho earlier in life influences regenerative and transcriptional responses later in life. If the positive attributes of Klotho, when applied to an "unhealthy" oldest-old environment, then become a negative stress, it is an idea that should be explored further.

We agree with the reviewer that this is a very interesting line of investigation and one of potentially great clinical relevance. Our team carefully weighed the feasibility of conducting such experiments. Unfortunately, considering the time to perform dose-testing pilot studies and power analysis as well as time required to age the animals following intervention, we estimated such studies would require almost one year. As an alternative, we have added text to the Discussion highlighting the value of such work for future studies (lines 431-433).

It would also be welcome to see how Klotho influences more translatable outcomes such as adaptability to exercise training, since the link between regenerative potential and age-associated sarcopenia is unclear.

To probe the effect of Klotho on exercise response, we evaluated the effects of AAV-Klotho administration on muscle function of mice following an exercise-induced injury (Armand, Launay et al. 2003). Briefly, 22-month-old male mice were randomized into one of two groups and treated with either AAV-Klotho or AAV-GFP via tail vein injection. The AAV-Klotho dose used was the same used for the severe injury protocol. Mice were acclimated to a treadmill for 4 days, given one day of rest, then underwent an eccentric exercise protocol, which consisted of running at 10 m/min for 60 minutes at a 15-degree decline. Though we could not find published studies that utilized this protocol in aged mice, the eccentric treadmill model has been shown to induce exercise-related tibialis anterior damage in young (2 months old) mice (Parise, McKinnell et al. 2008). We ensured that all mice included in the analysis were able to complete the protocol by assessing running performance according to the method established by Ríos-Kristjánsson (Ríos-Kristjánsson, Rizo-Roca et al. 2019). Animals performing under a threshold of 0.35 were removed from analyses (n=2 from each treatment group). One week after the treadmill protocol, an investigator blinded to the treatment group performed in vivo contractile testing of the TA muscle. We repeated this study in duplicate (two cohorts of mice, n=6/group/cohort).

We found that mice treated with AAV-Klotho had a slightly greater force production than those treated with AAV-GFP, though the result was not significant (p=0.11 as determined by a linear mixed effects model and adjusting for cohort). While these results suggest a potentially positive effect on adaptability to exercise for AAV-Klotho administration, a power analysis revealed that 28 mice per group would be needed to detect statistically significant differences between groups. Given that the effect size of the AAV-Klotho intervention is small (roughly 9% force improvement), we did not test additional animals as the initial results suggest that intervention at the dose tested is not likely to be physiologically relevant. These results are described on lines 237-250 and presented in Figure 5M and Figure 5—figure supplement 1.

19. For the mouse muscle IHC phenotyping, it is warranted to show other measures such as fiber type-specific CSA, quantification of ECM (via Sirius Red or Masson's), markers of denervation, as well as intramuscular adipose tissue infiltration (IMAT). Extending the analysis to a more oxidative/slow-twitch muscle such as the soleus would also be welcome, given that the TA mostly contains muscle fibers that aren't present in humans (Type IIB).

As suggested, we performed fiber-type staining of the TA muscle for each sex across age groups (Figure 1C, D, Figure 1—figure supplement 1C-G). Unfortunately, we did not have another muscle type, such as the soleus, available for quantification of fiber-type in a more oxidative muscle, and we did not have oldest-old mice available to repeat the protocol. However, our evaluation of the TA muscle across age groups revealed a shift in fiber characteristics and, specifically, a loss of the more oxidative TA type IIA and IIX phenotypes with age (Figure 1D). These findings are consistent with Augusto et al. and Giacomello et al. (2017, 2020).

We also quantified ECM using Masson’s trichrome staining. Results are presented in Figure 1D. Consistent with Collagen IV data, there was a significant increase in collagen deposition in the 27-29-month-old age groups of both sexes.

Additionally, we quantified both inter- and intramuscular adipose tissue using lipidtox staining, though no differences were observed according to age (Figure 1F, G).

Since we did not have tissues available for whole mount staining of the neuromuscular junctions available, as an alternative, we accessed the publicly available database of skeletal muscle transcriptome Tabula Muris Senis (Schaum, Lehallier et al. 2020) and quantified the change over time using well-established denervation genes *Musk*, *Ncam1*, and *Runx1* (Aare, Spendiff et al. 2016). We observed an increase with age for *Musk* and *Runx1* These data are presented in Figure 1—figure supplement 2. We also quantified these same genes in our own AAV-Kl treated tissues, with data presented in Figure 6—figure supplement 1C-D.

20. Further characterization of the AAV-Klotho regeneration experiments is warranted. Why is there less fibrosis in AAV-Kl? Are there fewer FAP/fibrogenic cells? Are there more satellite cells?

To answer this question, we quantified and compared the ratio of fibrogenic cells (FAPs) to muscle satellite cells (MuSCs) across experimental groups. Briefly, mice were treated with AAV-GFP or AAV-Klotho, as in other experiments (n=9 per group). Five days following injection, mice received a bilateral TAs were injured via cardiotoxin injection. Three days after injury, mice were euthanized to capture the muscle at the height of its fibrotic response. TA muscles were then digested, and cells were sorted via flow cytometry. We observed a twenty-fold decrease in the ratio of FAPs:MuSCs with Klotho treatment, which we hypothesize contributes to the decreased fibrosis observed in histological sections. This data is presented in Figure 5F.

Presenting collagen 4 intensity (Figure 4H) seems unorthodox. Please show Sirius Red or Masson's and normalized to area.

As suggested, we performed Masson’s Trichrome staining to confirm Collagen IV findings (Figure 5E).

In the aged AAV-Kl experiments, why is there less lipid accumulation, which appears to be specifically within (and not between) the muscle fibers? Was lipid accumulation between muscle fibers (IMAT) different? An Oil Red O analysis is warranted.

We have now quantified inter- and intramuscular lipid accumulation for these experiments using ImageJ to distinguish IMAT lipid from intramuscular lipid (for detailed see methods lines 518-526). Intramuscular, but not intermuscular, lipid accumulation was significantly decreased in old female mice treated with AAV-Kl. Conversely, in oldest-old female mice, we found that intramuscular lipid *increased* with AAV-Kl treatment, while intermuscular lipid was unchanged. The results are presented in Figure 6C, G, M. The observation of increased intramuscular lipid in oldest-old mice treated with AAV-Kl is likely a reflection of myofiber metabolic changes (Johannsen, Conley et al. 2012), which is consistent with our findings in Table 2. We now address this in the discussion on lines 417-425.

We used LipidTox staining based-analysis instead of Oil Red O staining because LipidTox has been shown to be more specific to native lipid droplets in fixed tissue (Fam, Klymchenko et al. 2018). Whereas Oil Red O staining has been used as a gold standard for decades, due to its poor solubility in solutions other than ethanol or isopropanol, the abrasive staining procedure can disrupt the native lipid droplets in fixed tissues (Fam, Klymchenko et al. 2018).

21. Serum levels of cholesterol and insulin of young mice (3-6 month) is necessary in Figure 4D to compare with those of old mice overexpressing Klotho.

We now present data showing levels of cholesterol, insulin, and glucose in young mice (Figure 4G, H, I), in addition to the other metabolites presented in the paper (Figure 4—figure supplement 1).

22. As previously reported by the authors in Nature communication ( 2018), it would be interesting to compare the Klotho's function in the mitochondria of muscle progenitor cell, with the Klotho's function here identified.

To address this comment, we performed TEM on muscles across intervention groups, and mitochondrial integrity was analyzed by an investigator blinded to the treatment groups using a method as previously described (Owen, Patel et al. 2019). Our findings reveal that the percent of mitochondria with normal structure is increased in mice treated with Klotho. Results are presented in Figure 5G, H.

23. To verify the author's suggestion in Discussion section, it is important to examine whether F2 or Kng2 plays a bifurcation role in old and oldest muscles.

We have removed the statement about F2 and Kng2 in the discussion. Instead, we focus on FGF signaling as a mediator of Klotho treatment in old and oldest-old animals.

24. The key experiment that seems to be missing is the manipulation of Klotho earlier in life to try and prevent the onset of "molecular disorder" later in life, especially since acute Klotho over-expression in the oldest old seemed to have the opposite effect.

As described above, we agree that these would be very interesting studies to conduct, but the longitudinal nature of these experiments precludes their execution as a part of this current work.

25. A summary figure that highlight the main findings of this manuscript would be a welcome addition.

Thank you for this suggestion. We have added a graphical abstract summarizing the major take-home messages of the work (Figure 8).

26. Abstract has no conclusions and does not properly reflect contents of the results.

We agree that our manuscript presents an abundant data set that is not fully captured in the abstract. With the 150-word limit, we have attempted to highlight and explain the major takeaway messages, and most notably, the novel use of network entropy to characterize integrative transcriptomic changes over time as well as the dependency of Klotho overexpression efficacy to reverse a sarcopenic phenotype according to age. We have revised the abstract in the hopes of making these main messages even more clear and direct.